# The role of Imp and Syp RNA-binding proteins in precise neuronal elimination by apoptosis through the regulation of transcription factors

**Wenyue Guan[1][†], Ziyan Nie[1][†], Anne Laurençon[1], Mathilde Bouchet[1], Christophe Godin[2], Chérif Kabir[1], Aurelien Darnas[1], Jonathan Enriquez[1]\***

[1]Institut de Génomique Fonctionnelle de Lyon, ENS de Lyon, CNRS, Univ Lyon 1, Lyon, France; [2]Laboratoire Reproduction et Développement des Plantes, ENS de Lyon, Lyon, France

**\*For correspondence:**
jonathan.enriquez@ens-lyon.fr

[†]These authors contributed equally to this work

**Competing interest:** The authors declare that no competing interests exist.

**Abstract** Neuronal stem cells generate a limited and consistent number of neuronal progenies, each possessing distinct morphologies and functions, which are crucial for optimal brain function. Our study focused on a neuroblast (NB) lineage in *Drosophila* known as Lin A/15, which generates motoneurons (MNs) and glia. Intriguingly, Lin A/15 NB dedicates 40% of its time to producing immature MNs (iMNs) that are subsequently eliminated through apoptosis. Two RNA-binding proteins, Imp and Syp, play crucial roles in this process. Imp+ MNs survive, while Imp−, Syp+ MNs undergo apoptosis. Genetic experiments show that Imp promotes survival, whereas Syp promotes cell death in iMNs. Late-born MNs, which fail to express a functional code of transcription factors (mTFs) that control their morphological fate, are subject to elimination. Manipulating the expression of Imp and Syp in Lin A/15 NB and progeny leads to a shift of TF code in late-born MNs toward that of early-born MNs, and their survival. Additionally, introducing the TF code of early-born MNs into late-born MNs also promoted their survival. These findings demonstrate that the differential expression of Imp and Syp in iMNs links precise neuronal generation and distinct identities through the regulation of mTFs. Both Imp and Syp are conserved in vertebrates, suggesting that they play a fundamental role in precise neurogenesis across species.

## eLife assessment

Guan and colleagues present **solid** arguments to address the question of how a single neural stem cell produces a defined number of progeny, and what influences its decommissioning. The focus of the experiments are two well-studied RNA-binding proteins: Imp and Syp. This is **valuable** work that will be of interest to the scientific community.

## Introduction

The central nervous system (CNS) receives information from the periphery, records and processes it to control different types of behavior such as communication or locomotion. This complex system relies on a network of neurons and glial cells, primarily generated during development by neuronal stem cells, to fulfill these functions. Each neuronal stem cell produces a specific number of neurons and glia with diverse identities at the appropriate time and location. The precise regulation of this process is essential, as any disruption in the molecular machinery controlling it can result in severe brain disorders or cancers. Notably, alterations in adult neurogenesis in humans

can contribute to psychiatric disorders such as schizophrenia (*Allen et al., 2016*; *Mahar et al., 2014*) and autism (*Hazlett et al., 2017*; *Wegiel et al., 2010*) or neurodegenerative diseases such as Alzheimer (*Gallardo, 2019*). Disruptions in neurodevelopmental programs are often observed in brain tumors, and tumors that arise in early childhood may be attributed to a dysregulation in the molecular machinery governing the termination of neurogenesis (*Jessa et al., 2019*). Elucidating the precise mechanisms that regulate the appropriate number of neuronal and glial cells is not only crucial for understanding biological system development and the origins of human diseases, but also for unraveling the evolution of the CNS's architecture and function. Recent studies in insects have indicated that variations in the number of neurons contribute to changes in neuronal circuits and behaviors (*Pop et al., 2020*; *Prieto-Godino et al., 2020*). In our research using *Drosophila*, we have identified a novel mechanism involved in controlling the accurate production of neurons during development.

In *Drosophila*, adult neurons and glia are mostly produced during larval and pupal stages by stem cells called neuroblast (NB). Similar to vertebrates, NBs undergo asymmetric divisions to self-renew and generate neuronal and glial progenies either directly or indirectly. In *Drosophila*, two major types of NBs are responsible for producing the majority of adult neurons and glia (*Harding and White, 2018*; *Homem et al., 2015*). Type I NBs generate ganglion mother cells (GMCs) that divide once to produce neurons and glia (*Doe et al., 1985*; *Karcavich and Doe, 2005*) while type II NBs generate intermediate neuronal progenitors that undergo multiple divisions to generate glia and/or neurons (*Homem et al., 2015*; *Bello et al., 2008*; *Bowman et al., 2008*). Under normal conditions, each NB in *Drosophila* produces a consistent number of mature neurons, although this number can vary between different NBs. The precise generation of a stereotypical number of progenies depends on three key parameters: the speed of NB division, the timing of NB neurogenesis termination, and programmed cell death (PCD) during the asymmetric division of GMCs. Various molecular mechanisms have been identified to regulate these parameters.

Most type I and II NBs end neurogenesis during the early pupal stages either by accumulating the transcription factor (TF) Prospero, triggering symmetrical division and resulting in two postmitotic cells (*Homem et al., 2014*; *Maurange et al., 2008*) or through a combination of autophagy and PCD (*Pahl et al., 2019*; *Siegrist et al., 2010*). Prior to the termination of neurogenesis, all NBs experience a decrease in size (*Homem et al., 2014*; *Maurange et al., 2008*). The reduction in size of NBs ending neurogenesis early in pupal stages is linked to a metabolic switch that enhances oxidative phosphorylation, leading to a terminal differentiation division (*Homem et al., 2014*). Conversely, the decrease in size observed in NBs terminating neurogenesis later, such as mushroom body NBs, correlates with a reduction in the activity of phosphatidylinositol 3-kinase, which acts as an autophagy inhibitor (*Homem et al., 2014*; *Pahl et al., 2019*). The process of terminating NB neurogenesis through autophagic cell death or terminal differentiation is commonly referred to as decommissioning. The temporal control of decommissioning is influenced by extrinsic signals like the steroid hormone ecdysone (*Homem et al., 2014*; *Pahl et al., 2019*) and as well as an intrinsic program characterized by the sequential expression of temporal RNA-binding proteins (RBPs) such as Imp (insulin-like growth factor 2 (IGF2) messenger RNA (mRNA)-binding proteins) and Syp (Syncrip) expressed in opposite temporal gradients within the NB, or through a temporal cascade of TFs. For instance, mushroom body NBs decommission during the late pupal stage due to a prolonged expression of Imp compared to other NBs (*Yang et al., 2017*). In the ventral nerve cord (VNC, analogous to our spinal cord), the timing of NB decommissioning is determined by temporal TFs, leading to apoptosis in type II NBs or terminal division in type 1 NBs (*Maurange et al., 2008*). Lastly, the decommissioning of NBs is also spatially regulated by spatial selectors such as Hox genes. For example, NB5-6 in thoracic segments produces a greater number of neurons compared to NB5-6 in the abdomen, due to the absence of abdominal Hox genes (*Karlsson et al., 2010*).

The rate of NB division is a molecularly controlled parameter that can influence the number of cells generated by a lineage. On average, NB divisions in larvae occur at a speed of approximately 80–90 min per division. However, this speed varies among different NBs (*Hailstone et al., 2020*). The heterogeneity in NB division speed appears to be regulated by the opposing temporal gradients of Imp and Syp within the NB. Imp promotes high-speed division by stabilizing *myc* RNA and increasing Myc protein levels, while Syp has an inhibitory effect on NB division speed by directly inhibiting Imp as development progresses (*Samuels et al., 2020*).

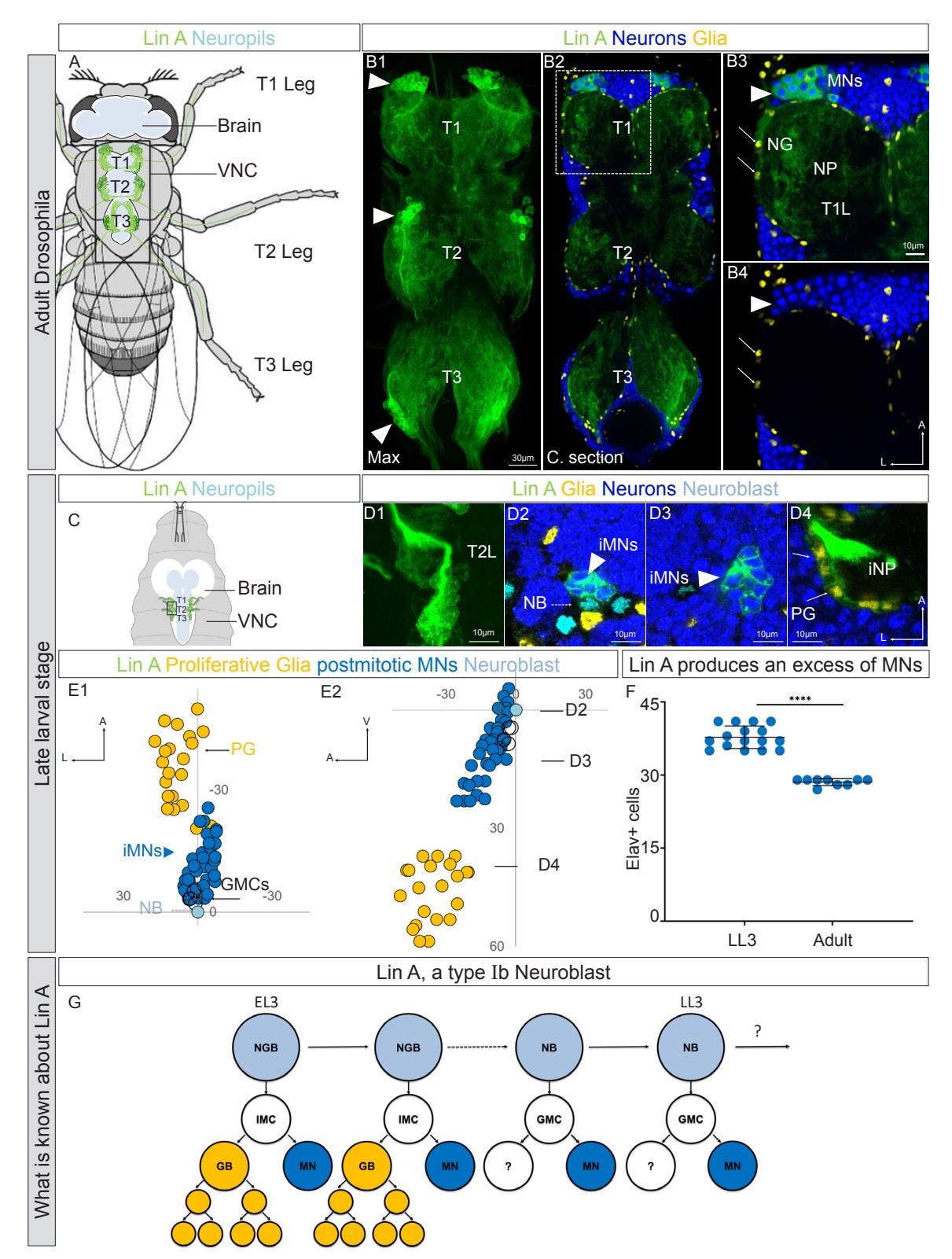

**Figure 1.** Lin A/15 as model to study how a stereotyped number of neurons is produced by an NB. (**A**) Drawing of an adult fly showing the position of the central nervous system (CNS) (white (cortex), blue (neuropiles)) and Lin A/15 leg MNs (green cell bodies and dendrites in the VNC and axons in the legs). Black box indicates the VNC imaged in (**B**). VNC: ventral nerve cord. (**B1**) Maximum projection of confocal sections of an adult VNC where the six Lin A/15s are genetically labeled with mCD8::GFP (green). (**B2**) Confocal section of the VNC in (**B1**) immunostained with anti-Elav (neuronal marker,

*Figure 1 continued on next page*

*Figure 1 continued*

blue) and anti-Repo (glia marker, yellow). T1, T2, and T3 indicate the Prothoracic, Mesothoracic, and Metathoracic neuromere, respectively. (**B3, B4**) Confocal section of first left prothoracic neuromere (T1L) (the boxed region in **B2**), arrowheads and arrows indicate Lin A/15 MN and glia cell bodies, respectively. NG: neuropile glia; NP: neuropile; MN: motor neuron. (**C**) Drawing of the anterior region of a third instar larva showing the position of the CNS (white (cortex) blue (neuropiles)) and immature Lin A/15 leg MNs (green). (**D1**) Maximum projection of confocal sections of the second left thoracic hemisegment (T2L) where Lin A/15 is genetically labeled with mCD8::GFP (green). (**D2–D4**) Confocal section of the second left thoracic hemisegment (T2L) in (**D1**) immunostained with anti-Elav (neuronal marker, blue), anti-Dpn (NB marker, cyan), and anti-Repo (glia marker, yellow). Arrowheads, doted arrows and arrows indicate immature Lin A/15 MNs (iMNs), Lin NB and Lin A/15 proliferative glia (PG), respectively. iNP: immature neuropile. (**E1, E2**) Plots of the relative position of each Lin A/15 cell from two perspectives: E1 ventral view, E2 lateral view. Axes: Anterior (A), Lateral (L), Ventral (V). Lin A proliferative glia (PG) are in yellow, Lin A/15 iMNs are in blue, Lin A/15 GMCs are in white and Lin A/15 NB is in cyan. Arrows indicate the positions of the confocal sections in (**D2–D4**). (**F**) Graph of the number of Elav+ MNs in a late third instar larva (LL3) versus that in an adult fly. Error bars represent standard deviations. Student's t test was performed to compare the difference between indicated groups. ****, P ≤ 0.0001. (**G**) Schematic of the Lin A/15 type Ib division. NGB: neuroglioblast, NB: neuroblast, IMC: intermediate mother cell, GMC: ganglion mother cell, GB: glioblast, MN: motoneuron. Note 1: The destiny of the MN sister cell during the second phase of division is unknown. Note 2: Lin A/15 development has not been studied during pupal stages.

Furthermore, PCD after asymmetric division of the GMC acts as another influential factor in shaping the final clonal size of each lineage. Between 40% and 50% of hemi-lineages derived from type I NBs undergo PCD (*Kumar et al., 2009*; *Truman et al., 2010*). Interestingly, when PCD is blocked in the midline NBs of the VNC, the 'undead' neurons differentiate and form complex and functional arborizations (*Pop et al., 2020*). Recent studies have demonstrated how variation in PCD patterns between different insect species could change their behavior suggesting that probably all mechanisms controlling the number of neurons potentially play a role during the behavior evolution (*Pop et al., 2020*; *Prieto-Godino et al., 2020*).

The diverse characteristics of neural stem cells (NBs), such as their proliferation termination, proliferation rate, and the generation of a GMC producing a dying cells by asymmetric division, may account for the varying number of neurons produced by individual NBs. Here, we have discovered that the fate of immature neurons also plays a crucial role in shaping the final number of neurons produced by a single stem cell.

In this study, we investigated a specific *Drosophila* lineage known as Lin A (also called Lin 15), which gives rise to 29 adult motoneurons (MNs) per ganglion, responsible for innervating leg muscles during the adult stage. This lineage also produces most of the astrocytes and ensheathing glia of the thoracic ganglion (*Baek and Mann, 2009*; *Brierley et al., 2012*; *Enriquez et al., 2018*, *Figure 1A–B4*). The process of division in Lin A/15 NB during larval stages follows a particular pattern termed type Ib (*Figure 1C–G*). Initially, Lin A/15 NB generates intermediate mother cells (IMCs) that give rise to a postmitotic cell and a proliferative glioblast (*Figure 1G*). Subsequently, during a second phase, Lin A/15 switches to a classical type I division mode, producing only postmitotic MNs (*Figure 1G*, *Enriquez et al., 2018*). As development progresses, an adult Lin A/15 consists of a predetermined number of MNs (*Figure 1F*). Interestingly, during larval stages, Lin A/15 NB produces an excess of neurons, which are selectively eliminated before reaching the adult stage (*Figure 1F*).

We first established that after producing glia and MNs, Lin A/15 GMCs produce one postmitotic MN and a sibling cell that is eliminated by PCD shortly after birth. This mode of division continues until the decommissioning of Lin A/15 NB by PCD, resulting in the production of supernumerary MNs. The excess MNs are then precisely and progressively eliminated by PCD from early pupal stages until the end of Lin A/15 neurogenesis, eventually reaching the final number of 29 MNs. Both the decommissioned Lin A/15 NB and the MNs eliminated by PCD are characterized by being Imp− and Syp+. Through genetic manipulations, we discovered that altering the temporal and spatial expression patterns of Imp and Syp can change the timing of NB decommissioning and the number of immature MNs (iMNs) that survive.

Further analysis of the expression patterns of various TFs published on our previous work (*Guan et al., 2022c*), using the computational tool PCCD 2.0, revealed that the last-born MNs exhibit a distinct set of TFs compared to the first-born MNs. Moreover, we found that changes in the expression levels of Imp and Syp directly affect the expression of at least three TFs: Nvy, Jim, and RunxA. Overexpression of Imp or knocking down Syp induces Jim expression and downregulates RunxA and Nvy in the last-born surviving MNs. This regulation of TF codes by Imp and Syp occurs at least in MNs 29–34. Notably, the new set of TFs induced in the last-born MNs closely resembles the TF

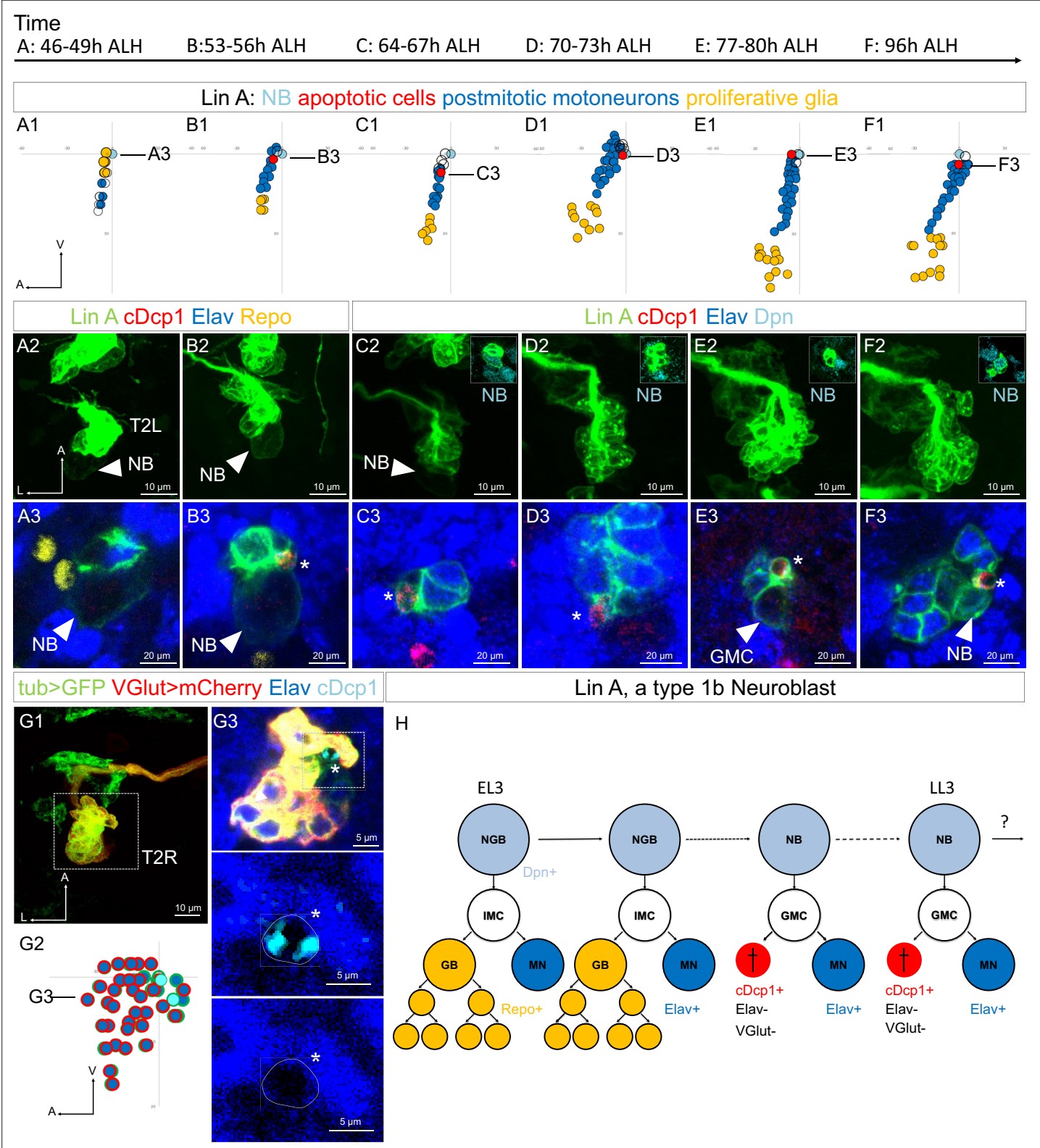

**Figure 2.** The MN sibling cells die through programmed cell death (PCD) during the second phase of Lin A/15 NB division. (**A1–F3**) Graphs and confocal images showing the development of Lin A/15 during larval stages, the developmental time points are indicated on top. (**A1, B1, C1, D1, E1, F1**): Graphs of the relative position of each Lin A/15 cell in (**A3, B3, C3, D3, E3, F3**) from a lateral perspective. Axes: Anterior (A), Ventral (V). Lin A/15 proliferative glia are in yellow, Lin A/15 immature MNs are in blue, Lin A/15 GMCs are in white, Lin A/15 NB is in cyan, Lin A/15 cDcp1+ cells are

*Figure 2 continued on next page*

*Figure 2 continued*

in red. The black lines indicate the positions of the confocal section in (**A3–F3**). (**A2, B2, C2, D2, E2, F2**): Maximum projection of confocal sections of the second left thoracic hemisegment (T2L) where Lin A/15 is genetically labeled with mCD8::GFP (green). The boxes in (**C2–F2**) are confocal sections showing Lin A/15 NB immunostained with anti-Dpn (cyan). Note: In A2–B2, the NB is easily recognizable by its size (arrowheads). (**A3, B3, C3, D3, E3, F3**): are magnified confocal sections of samples in (**A2, B2, C2, D2, E2, F2**) immunostained with anti-cDcp1(red), anti-Elav (neuronal marker, blue) and anti-Repo (glia marker, yellow) (**A2, B2**) or with anti-Dpn (NB marker, cyan) (**C2, D2, E2, F2**). Asterisk in (**B3, C3, D3, E3, F3**) indicate of the cDcp1+ Elav− apoptotic cell. (**G1**) Maximum projection of confocal sections of a second right thoracic hemisegment (T2R) with a Lin A/15 MARCM clone genetically labeled with mCD8::GFP (green) under the control of *tub-Gal4* and mCherry (red) under the control of *VGlut-LexA::GAD*. (**G3**) Confocal section of the second right thoracic hemisegment (T2R) (boxed region in G1) immunostained with anti-Elav (blue) and anti-cDcp1 (cyan). The arrowheads indicate the cDcp1+ Elav− *VGlut*− apoptotic cell. (**G2**) Graph of the relative position of each Lin A/15 cell (excluding the proliferative glia) in (**G1**) from a lateral perspective. Axes: Anterior (A), Lateral (L), Ventral (V). Lin A/15 immature MNs are in blue (Elav+), the blue cells surrounded in red are the GFP+ Elav+ VGlut+ immature MNs, the blue cells surrounded in green are the GFP+ Elav+ VGlut− immature MNs (last-born MNs) and the cyan cells surrounded in green are the cDcp1+ GFP+ VGlut− Elav− apoptotic cell. Note: The NB (white cell surrounded in green) has been identified by its size. The black line indicates the position of the confocal section in (**G2**). (**G3**) Confocal section of the second right thoracic hemisegment (T2R) (boxed region in G1) immunostained with anti-Elav (blue) and anti-cDcp1 (cyan). The arrowheads indicate the cDcp1+ Elav− *VGlut*− apoptotic cell. (**H**) Schematic of the Lin A/15 type Ib division. NGB: neuroglioblast, NB: neuroblast, IMC: intermediate mother cell, GMC: ganglion mother cell, GB: glioblast, MN: motoneuron. The markers used to label each type of Lin A cells are indicated.

code seen in earlier-born MNs (MNs 16–24). To further validate our findings, we genetically imposed a code found in young MNs (MNs 16–24) onto the surviving last-born MNs by overexpressing Jim and suppressing Nvy function. This manipulation resulted in the survival of the last-born MNs. These results suggest that the last-born MNs undergo apoptosis due to their failure to express a functional TF code, and this code is post-transcriptionally regulated by the opposite expression of Imp and Syp in iMNs.

Previous studies and our work cements the role of Imp and Syp as two multitasking proteins that can modulate the number of neuronal cells through different mechanisms: the timing NB decommissioning (*Yang et al., 2017*) and NB division speed (*Samuels et al., 2020*) and the number of MNs surviving (this work).

## Results

## Lin A/15 switches to a classical type I division during larval stages where the GMC produces one MN and a dying cell

Lin A/15 produces MNs and glia during the first phase of NB division and only MNs during a second phase. It has been suggested that during the second phase Lin A/15 GMCs could produce an MN and an apoptotic sibling cell (*Truman et al., 2010*).

We precisely characterized Lin A/15 development by labeling it with GFP during the first and the second phase of NB division by using a Lin A/15 tracing system (*Awasaki et al., 2014*; *Lacin and Truman, 2016*) and immunostained it against a cleaved form of the *Drosophila* caspase-1 (cDcp1), an apoptotic marker (*Figure 2*). During the first phase, 40–49 hr after larval hatching (ALH), no cDcp1+ cells were observed, confirming our previous study that during early developmental stages Lin A/15 NB produces an IMC that divides once to produce a glioblast and a postmitotic MN (*Figure 2A1–A3*, *Enriquez et al., 2018*). During the second phase, 53–96 hr ALH, apoptotic cells cDcp1+ Elav were detected close to Dpn+ Lin A/15 NB (Dpn and Elav are NB and postmitotic neuronal markers, respectively), suggesting that PCD occurs soon after asymmetric division. The absence of Elav in the apoptotic cells also suggested that the sibling cells die before acquiring a neuronal identity (*Figure 2B1–F3*). Consistent with this idea, we then generated Lin A/15 MARCM clones to label all Lin A/15 cells with GFP under the control of a *tub-Gal4* transgene, and *VGlut+* MNs with mCherry by using *VGlut-LexA::GAD*, a gene trap transgene of the gene coding for the vesicular glutamate transporter expressed by all *Drosophila* MNs. We dissected late third instar larva (LL3) and observed that cDcp1+ Elav cells did not express *VGlut* (*Figure 2G1–G3*).

Together, our results show that Lin A/15 GMCs produce an MN and a sibling cell eliminated by PCD during the second phase of Lin A division (*Figure 2H*). Moreover, PCD of the sibling cell occurs soon after birth before they express neuronal marker such as Elav or *VGlut*.

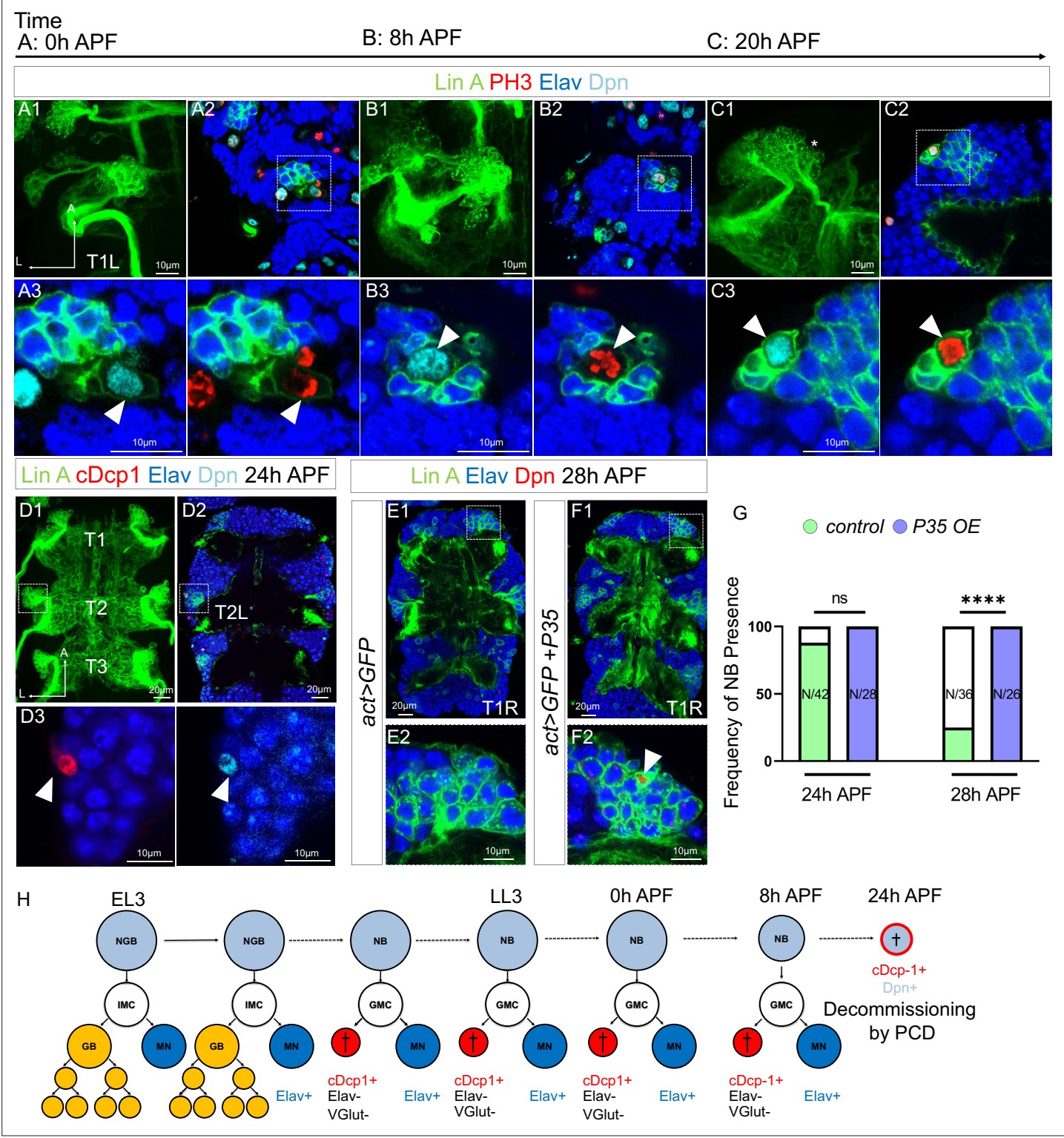

**Figure 3.** Lin A/15 NB decommissions at 24 hr APF through programmed cell death (PCD). (**A1–C3**) Confocal images showing the development of Lin A/15 during pupal stages, the developmental time points are indicated on top. APF: after pupa formation. (**A1, B1, C1**) Maximum projection of confocal sections of the first left thoracic hemisegment (T1L) where Lin A/15 is genetically labeled with mCD8::GFP (green). (**A2, B2, C2**) Confocal sections showing Lin A/15 immunostained with anti-Elav (blue), anti-Dpn (cyan), and anti-pH3 (red, phospho-Histone3), mitosis-specific marker (**A3, B3, C3**) magnifications of the boxed region in (**A2, B2, C2**). Arrowheads indicate the proliferative Lin A/15 NBs (Dpn+ pH3+). (**D1**) Maximum projection of confocal sections of three thoracic ganglions (T1, T2, T3) at 24 hr APF where all six Lin A/15s are genetically labeled with mCD8::GFP (green). (**D2**)

*Figure 3 continued on next page*

*Figure 3 continued*

Confocal section of thoracic ganglions in (**D1**) immunostained with anti-Elav (blue), anti-Dpn (cyan), and anti-cDcp1 (red). (**D2**) Magnifications of the boxed region in (**D2**) (left Mesothoracic neuromere, T2L). Arrowheads indicates the apoptotic Lin A/15 NB (Dpn+ cDcp1+). (**E1–F1**) Confocal images showing the absence vs presence of Lin A/15 NB at 28 hr APF in Control vs P35 overexpression (OE) conditions. Six Lin A/15s are genetically labeled with mCD8::GFP (green), Lin A/15 NB and MNs are visualized with anti-Dpn (red) and anti-Elav (blue), respectively. (**E2–F2**) Magnifications of the boxed region in (**E1–F1**) (first right thoracic hemisegment, T1R) indicate the presence of NB (Dpn+, arrowhead) in P35 OE condition. (**G**) Graph of the frequency of NB presence (number of Lin A/15 samples analyzed is indicated within each bar) at different developmental time points under different genetic conditions: absence of NB (white), NB presence in *Control* (green), NB presence in *P35 OE* (purple). Student's t test was performed to compare the difference between indicated groups. ns, P > 0.05, considered not significant; ****, P ≤ 0.0001. (**H**) Schematic of the Lin A/15 type Ib division during larval and pupal stages. NGB: neuroglioblast, NB: neuroblast, IMC: intermediate mother cell, GMC: ganglion mother cell, GB: glioblast, MN: motoneuron. The markers used to label each type of Lin A/15 cells are indicated.

The online version of this article includes the following figure supplement(s) for figure 3:

**Figure supplement 1.** The motoneuron (MN) sibling cells die through programmed cell death (PCD) during the second phase of Lin A/15 neuroblast (NB) division.

**Figure supplement 2.** The volume of the Lin A/15 neuroblast (NB) continuously decrease during development.

**Figure supplement 3.** Lin A/15 neuroblast (NB) does not enter into autophagy.

## Lin A/15 NB decommissions 24 hr after pupa formation through PCD

We subsequently explored the development of Lin A/15 during the pupal stages to determine when the NB stops producing MNs and terminates neurogenesis. We genetically labeled Lin A/15 with GFP and performed immunostaining against Dpn to label Lin A/15 NB and against phospho-Histone H3 (PH3), a marker of cell proliferation. We revealed that Lin A/15 NB (PH3+/Dpn+) continues proliferating from 0 hr after pupa formation (APF) until 20 hr APF (*Figure 3A1–C3*). During early pupal stages, cDcp1+Elav− cells were also detected, suggesting that Lin A/15 NB keeps producing a GMC that divides once into an MN and an apoptotic cell (*Figure 3—figure supplement 1*).

At 24 hr APF, Lin A/15 NB is no longer detected in most of our samples (*N* = 4/17, number of Lin A/15 with an NB), revealing the end of Lin A/15 neurogenesis at this stage. Similar to what has been shown for other NBs, the volume of Lin A/15 NB decreases throughout development until the termination of Lin A/15 neurogenesis (*Figure 3—figure supplement 2*). At 24 hr APF, the remaining Lin A/15 NBs are extremely small and express cDcp1, suggesting a decommissioning of Lin A/15 NB through PCD (*Figure 3D*). To confirm this result, we then inhibited PCD in Lin A/15 by using the baculovirus P35 protein, an inhibitor of apoptosis in insects (*Hay et al., 1994*). Under these experimental conditions, Lin A/15 NB survived at least until 28 hr APF (*Figure 3E–G*). Furthermore, we did not detect autophagic markers in the NB, such as Atg8 and LysoTracker (*Mauvezin et al., 2014*) suggesting that the PCD of the NB is not linked to autophagy such as the mushroom body NBs (*Figure 3—figure supplement 3*).

Our results reveal that Lin A/15 NB continues producing MNs during early pupal stages and, unlike most NBs described in the thoracic segments of the VNC (*Maurange et al., 2008*), Lin A/15 NB terminates its proliferative phase through PCD at 24 hr APF in all thoracic ganglia (*Figure 3H*).

## Opposite temporal gradients of Imp and Syp control the timing of Lin A/15 NB decommissioning

We conducted further investigations to determine whether Imp and Syp could regulate the timing of Lin A/15 NB decommissioning.

During Lin A/15 neurogenesis, we carefully examined the expression pattern of Imp and Syp in Lin A/15 NB. We employed two approaches to achieve this: smFISH to determine the absolute expression of *Imp* and *Syp* mRNA, and double immunostaining to quantify the relative expression of Imp and Syp proteins.

At early L3 stages (46–49 hr ALH), Imp protein and *Imp* RNA are highly expressed in Lin A/15 NB, while Syp protein and *Syp* RNA are not detectable (*Figure 4A, F, G, L*). As neurogenesis progresses to mid-L3 larval stages (70–73 hr ALH), Lin A/15 NB starts to express Syp protein and *Syp* RNA, and the expression of Imp begins to decrease (*Figure 4B, F, H, L*). Toward the end of the larval stages, there is a reversal in the expression pattern: Imp protein and *Imp* RNA are weakly expressed, while Syp protein and *Syp* RNA are highly expressed (*Figure 4C, F1, L*). This opposite expression pattern

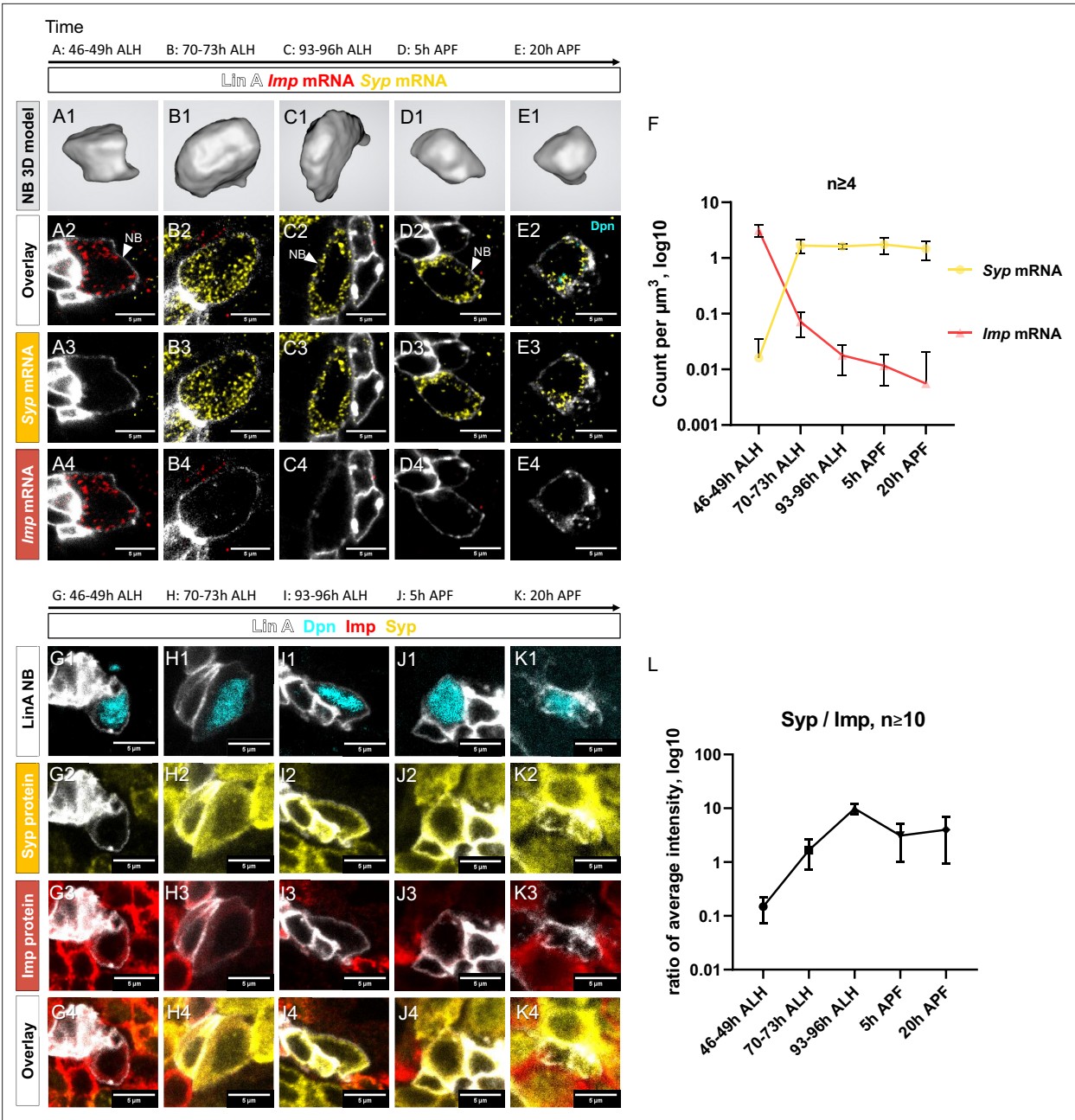

**Figure 4.** Opposite temporal expression of Imp and Syp in Lin A/15 neuroblast (NB). (**A1–F**) smFISH of *Imp* (red) and *Syp* (yellow) mRNA in Lin A/15 NB labeled with GFP (white) at different time points during development. (**A1, B1, C1, D1, E1**) 3D renderings of Lin A/15 NB segmentations used to quantify total numbers of *Imp* (red) and *Syp* (yellow) mRNA. (**A2–A4, B2–B4, C2–C4, D2–D4, E2–E4**) confocal sections. (**F**) Graph of *Imp* and *Syp* mRNA concentrations in Lin A/15 NB at different time points ($n \geq 4$ for each time point). Error bars represent standard deviations. Note: The Lin A/15 GFP+ NB is recognized based on its large size. Because the NB size decreases drastically in the pupal stage, smFISH against *dpn* (cyan) was performed at 20 hr APF to recognize Lin A/15 NB (**E2–E4**). (**G1–L**) Co-immunostaining of Imp (red), Syp (yellow), and Dpn (NB marker, cyan) protein in Lin A/15 NB labeled with GFP (white) at different time points during development. (**G1–K4**) Confocal sections. (**L**) Graph of relative expression levels of Imp and Syp protein in Lin A/15 NB at different time points, represented by relative ratios of staining intensity values measured in ImageJ ($n \geq 10$ for each time point). Error bars represent standard deviations. The developmental time points are indicated on the top of each panel. All scale bars are 5 μm.

The online version of this article includes the following source data for figure 4:

**Source data 1.** (4F) smFISH quantification: *Imp* and *Syp* RNA counts and neuroblast volumes.

**Source data 2.** (4L) Immunostaining quantification: Imp and Syp staining signals in neuroblasts.

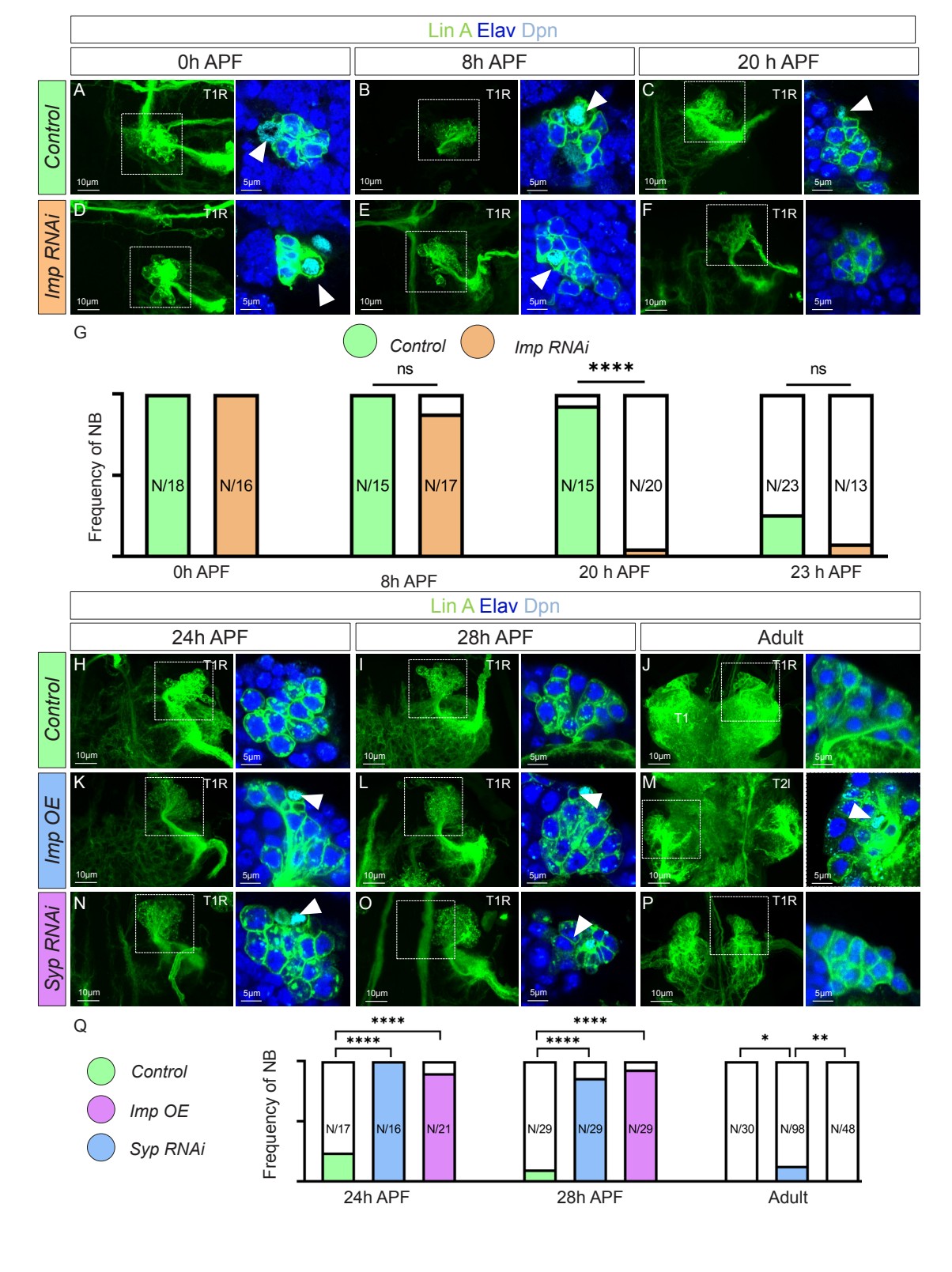

**Figure 5.** Opposite temporal expressions of Imp/Syp control the timing of Lin A/15 neuroblast (NB) decommissioning. Confocal images showing the development of *WT* (**A–C**), *Imp RNAi* (**D–F**), from pupal stages until the adult stage. Left: Maximum projection of confocal sections of Lin A/15 genetically labeled with mCD8::GFP (green); right: Confocal sections of the boxed regions immunostained with anti-Elav (neuronal marker, blue) and anti-Dpn (NB marker, cyan). The developmental time points are indicated on the top of each panel. Arrowheads indicate the presence of NB (Dpn+).

*Figure 5 continued on next page*

Figure 5 continued

(**G**) Graph of the frequency of NB presence (number of Lin A/15 samples analyzed is indicated on each bar) at different developmental time points under different genetic conditions: absence of NB (white), NB presence in *Control* (green), and NB presence in *Imp RNAi* (orange). Student's t test was performed to compare the difference between indicated groups. ns, P > 0.05, considered not significant. ****, P ≤ 0.0001. Confocal images showing the development of *WT* (**H–J**), *Imp OE* (**K–M**), and *Syp RNAi* Lin A/15 (**N–P**) from pupal stages until the adult stage. Left: Maximum projection of confocal sections of Lin A/15 genetically labeled with mCD8::GFP (green); right: Confocal sections of the boxed regions immunostained with anti-Elav (neuronal marker, blue) and anti-Dpn (NB marker, cyan). The developmental time points are indicated on the top of each panel. Arrowheads indicate the presence of NB (Dpn+). (**Q**) Graph of the frequency of NB presence (number of Lin A/15 samples analyzed is indicated on each bar) at different developmental time points under different genetic conditions: absence of NB (white), NB presence in *Control* (green), NB presence in *Imp RNAi* (orange), NB presence in *Imp OE* (blue), and NB presence in *Syp RNAi* (purple). Student's t test was performed to compare the difference between indicated groups. *, P ≤ 0.05; **, P ≤ 0.01; ****, P ≤ 0.0001.

of Imp and Syp becomes more pronounced during pupal stages, where Imp protein and *Imp* RNA are no longer detectable in the NB just before decommissioning (*Figure 4D–F, J–L*).

To investigate the impact of Imp and Syp on the lifespan of Lin A/15 NB, we conducted specific manipulations. Knocking down Imp in Lin A/15 NB resulted in premature decommissioning at 20 hr APF (*Figure 5A–G*). Conversely, prolonging the expression of Imp or knocking down Syp led to an extension of Lin A/15 NB's lifespan until the young adult stages and at least 28 hr after pupa formation (APF), respectively (*Figure 5H–Q*). Furthermore, we observed that the temporal expression of Imp/Syp controls the timing of Lin A/15 NB decommissioning similarly in all thoracic segments.

Our findings provide evidence that the sequential expression of Imp and Syp plays a crucial role in regulating the timing of Lin A/15 NB decommissioning, ultimately terminating neurogenesis (*Figure 5*).

## The last-born neurons produced by Lin A/15 are eliminated by PCD

The number of Lin A/15 MNs in adult flies is highly consistent and gradually established during development (*Figure 1F* and *Figure 6K*). At 0 hr APF, Lin A/15 reaches its maximum number of Elav+ neurons (*N* = 39, SD = 2, number of Lin A/15 Elav+ neurons) (*Figure 7K*). However, even though Lin A/15 NB continues to divide in early pupal stages (*Figure 3A–C*), the number of Elav+ neurons progressively decreases until reaching almost the final number of MNs at 24 hr APF (*N* = 32, SD = 1 at 24 hr APF compared to adult: *N* = 29, SD = 1) (*Figure 7K*).

To investigate if the supernumerary postmitotic MNs are eliminated by PCD, we genetically labeled Lin A/15 with GFP and performed immunostaining against Dpn, Elav, and cDcp1 during pupal stages (*Figure 6A1–C3*). Unlike in larval stages, we detected Elav+ cDcp1+ neurons close to the NB, indicating that the last-born Elav+ neurons are progressively eliminated by apoptosis from 0 to 24 hr APF (*Figure 6A1–C3*). To further confirm that the last-born MNs are eliminated by PCD, we fed L3 larvae with 5-ethynyl-2'deoxyuridine (Edu) to label only the late-born MNs and dissected the CNS at early pupal stage (5 hr APF) (*Figure 6D1–D7*). The first-born MNs were Edu– (*N* = 30, SD = 2, average number of Edu– MNs at 5 hr APF) and located away from the NB, while the last-born MNs (*N* = 7, SD = 2, number of Edu+ MNs at 5 hr APF) were Edu+ and positioned close to the NB (*Figure 6D5–D7*). Additionally, we found that cDcp1+ Elav+ neurons were always Edu+, and no cDcp1+ Edu– neurons were observed (*N* = 16, number of Lin A/15 analyzed). Further dissection of CNSs at late pupal stage (17 hr APF) showed a significant decrease in the number of Elav+ Edu+ MNs (*N* = 2, SD = 2, number of Edu+ MNs at 17 hr APF), confirming the progressive elimination of all these late-born MNs (*Figure 6D1–F*). Both results indicate that the last-born MNs undergo PCD during pupal stages.

To confirm that the supernumerary MNs are eliminated by PCD, we conducted an experiment to inhibit apoptosis in MNs by ectopically expressing the antiapoptotic gene P35. For this purpose, we employed the MARCM technique with the *VGlut-Gal4* (also known as OK371-Gal4) enhancer trap driver to express P35 specifically in the supernumerary immature neurons. It is important to note that this driver is exclusively expressed in Lin A/15 Elav+ cells (*Figure 2*). Consequently, the expression of VGlut>P35 is expected to fail in inhibiting apoptosis in the Elav– cells eliminated by PCD after the GMC division (Lin A/15 hemi-lineage). However, it should effectively inhibit apoptosis in the last-born Elav+ MNs during the pupal stage. As anticipated with this genetic manipulation, we observed similar numbers of neurons produced in L3 larvae in the Lin A/15 MARCM clone expressing P35 compared to the WT Lin A/15, while more Lin A MNs survived into adulthood (*Figure 6F–J*). These results provide further evidence supporting the notion that supernumerary MNs are indeed eliminated by PCD.

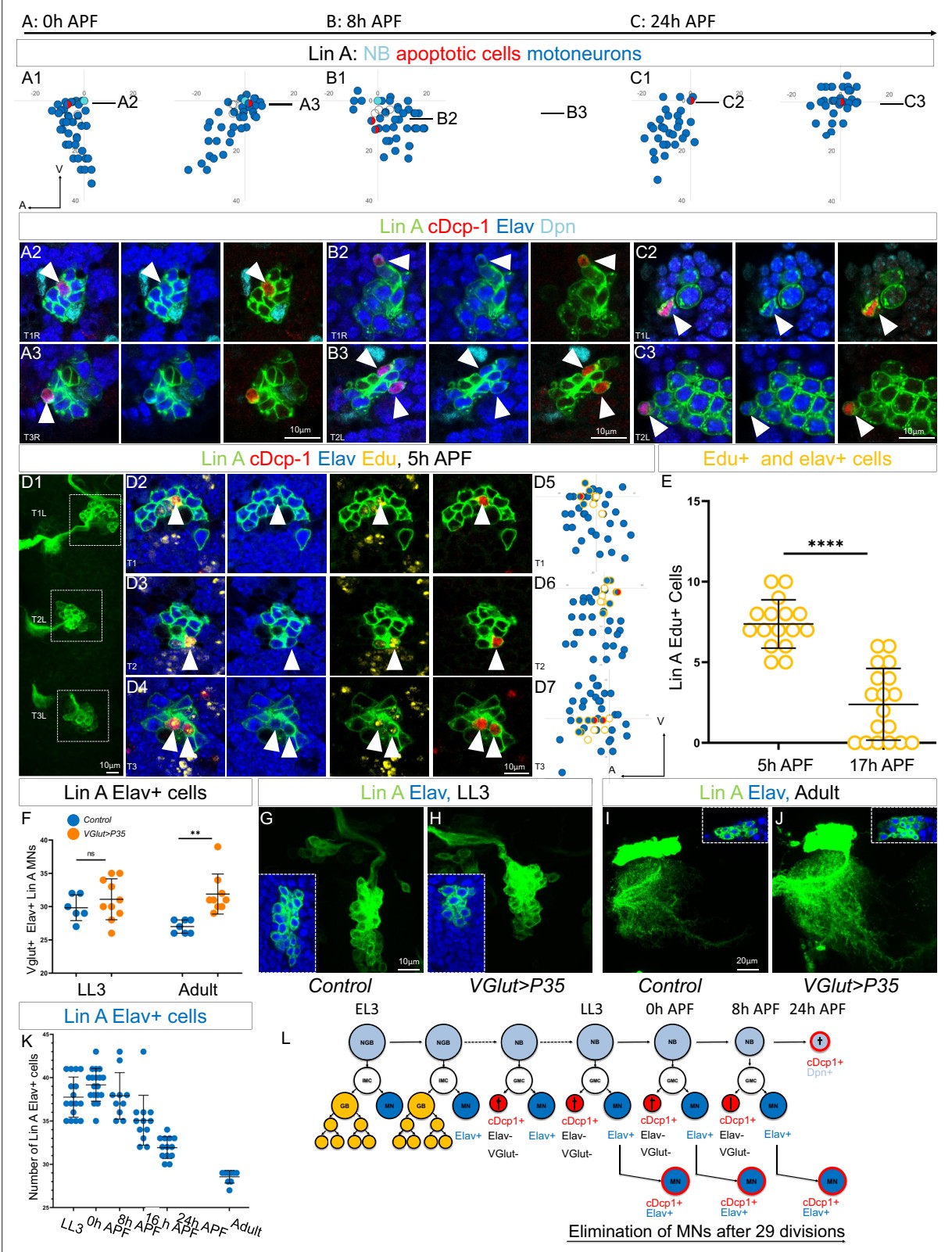

**Figure 6.** The last-born Lin A/15 MNs are eliminated by programmed cell death (PCD) during early pupal stages. (**A1–C3**) Graphs and confocal images showing the PCD pattern in Lin A/15 MNs during pupal stages, the developmental time points are indicated on top. (**A1, B1, C1**) Graphs of the relative position of each Lin A/15 cell of the boxed regions in (**A2, B2, C2**) from a lateral perspective. Axes: Anterior (A), Ventral (V). Lin A/15 immature MNs are in blue, Lin A/15 GMCs are in white, Lin A/15 NB is in cyan, and cDcp1+ MNs are in red and blue. The black lines indicate the positions of the

*Figure 6 continued on next page*

*Figure 6 continued*

confocal sections in (**A3, A4, B3, B4, C3, C4**). (**A2–C3**) Confocal sections of Lin A/15 genetically labeled with mCD8::GFP (green). The confocal sections showing the Elav+, cDcp1+ cells immunostained with anti-Dpn (cyan), anti-Elav (blue), and anti-cDcp1 (red) are indicated in (**A1, B1, C1**). Arrowheads indicate the apoptotic MNs (Elav+, cDcp1+). (**D1**) Maximum projection of confocal sections of three left thoracic hemisegments at 5 hr APF where Lin A/15 is genetically labeled with mCD8::GFP (green). (**D2, D3, D4**) Confocal sections of the three left thoracic hemisegments (boxed regions in (**E1**)) immunostained with anti-cDcp1(red), anti-Elav (neuronal marker, blue), and 5-ethynyl-2'deoxyuridine (Edu) (yellow). Arrowheads indicate the Elav+, cDcp1+ Edu+ cells. Note: the larvae were fed with Edu from 74 to 96 hr after larval hatching (ALH) to label only the last-born motoneurons with Edu (close to the NB). The cDcp1+ Elav+ cells are always Edu+. (**D5, D6, D7**) Graphs of the relative position of each Lin A/15 cell of the boxed regions in (**D1**) from a lateral perspective. Axes: Anterior (A), Ventral (V). Lin A/15 immature MNs are in blue, Lin A/15 GMCs are in white, Lin A/15 NB is in cyan, and Elav+, cDcp1+ cells are in red and blue, Edu+ cells are circled in yellow. Note: The last-born MNs and the GMCs as well as NB are all Edu+ (circled yellow) and the apoptotic Elav+ cDcp1+ cells are always part of this population of Edu+ cells demonstrating that the last-born MNs are dying. (**E**) Graph of the number of Edu+ Lin A/15 MNs at 5 and 17 hr APF, of which larvae are fed with Edu from 74 to 96 hr ALH. Note: The number of Edu+ cells neurons decrease significantly between 5 and 17 hr APF demonstrating that the last-born MNs are eliminated. n = 16. (**F**) Graph of the number of Elav+ *VGlut*+ Lin A/15 cells in third instar larvae (LL3) and adults (Adult) of *WT* versus *VGlut>P35* Lin A/15 MARCM clones. n ≥ 6. (**E,F**) Error bars represent standard deviations. Student's t test was performed to compare the difference between indicated groups. ns, P > 0.05, considered not significant; **, P ≤ 0.01; ****, P ≤ 0.0001.Maximum projection of confocal sections of the left T1 segment (T1L) of third instar larva (**G, H**) and the left prothoracic neuromere (T1L) in adult fly (**I, J**) containing a WT (**G, I**) or a *VGlut>P35* (**H,J**) Lin A/15 MARCM clone. Insets in (**G–J**) indicate the Elav+ GFP+ cells. (**K**) Graph of the number of Elav+ Lin A/15 neurons at different developmental time points. Error bars represent standard deviations. n ≥ 9. (**L**) Schematic of the Lin A/15 type Ib division during larval and pupal stages. NGB: neuroglioblast, NB: neuroblast, IMC: intermediate mother cell, GMC: ganglion mother cell, GB: glioblast, MN: motoneuron.

Overall, our results provide strong evidence that the elimination of supernumerary MNs by PCD is not random. Instead, only the last-born neurons are precisely and progressively eliminated through PCD during pupal stages from 0 to 24 hr APF (*Figure 6L*).

## Opposite spatial gradients of Imp and Syp in postmitotic neurons determine the pattern of PCD

The serially derived neuronal progenies of the NB express Imp or Syp according to the expression levels in the NB: Imp is highly abundant in first-born MNs, while Syp is highly present in late-born MNs (*Figure 4A–J*). This expression in postmitotic neurons has been proposed to be the consequence of the passive inheritance of these RBPs during NB and GMC division (*Mauvezin et al., 2014*). While this could be the case here, our previous studies revealed that during larval stages, Imp and Syp are also actively transcribed in postmitotic MNs (*Guan et al., 2022c*).

Although the mechanism of Imp/Syp expression in postmitotic neurons is not fully understood, we decided to quantify the relative expression of Imp and Syp in iMNs according to their birth order in LL3. In our previous work (*Guan et al., 2022c*), we revealed a link between the distance from the NB to immature neurons and their birth order in LL3. Using this parameter to identify older versus younger MNs, we found an opposite gradient of Imp and Syp expression according to birth order, with Imp expression progressively decreasing and Syp expression progressively increasing in older MNs (*Figure 7*). Based on this observation, we challenged the hypothesis that this gradient is responsible for controlling the precise elimination of last-born MNs.

Notably, during pupal stages, the last-born Lin A/15 MNs that are eliminated by PCD are Imp− and Syp+ (*Figure 8A1–A5*). These observations suggest that the distinct expression patterns of these two RBPs in postmitotic neurons may play a crucial role in determining the final number of adult MNs surviving.

Our genetic tools not only enable us to study the decommissioning of Lin A/15 NB under various genetic conditions but also to precisely investigate the fate of its progeny. Knocking down Imp in Lin A/15 resulted in premature PCD of Elav+ MNs during larval stages (*Figure 8C1–D3*) and an increase in the number of Elav+ neurons eliminated by PCD during pupal stages (*Figure 8B1*). Conversely, by prolonging the expression of Imp or knocking down Syp in Lin A/15 NB and its progeny, we were able to inhibit PCD of the supernumerary neurons (*Figure 8B2, E1–J3*). These genetic experiments, which manipulated the levels of Imp and Syp in the NB and its postmitotic progeny, led to changes in the final number of neurons produced by Lin A/15 (*Figure 8K*) by altering the timing of NB decommissioning and the pattern of PCD in postmitotic neurons. The modulation of PCD pattern in postmitotic neurons by Imp and Syp could result from an autonomous function in postmitotic MNs or a change in the temporal identity of Lin A/15 NB. To separate their functions in the NB and MNs, we ectopically

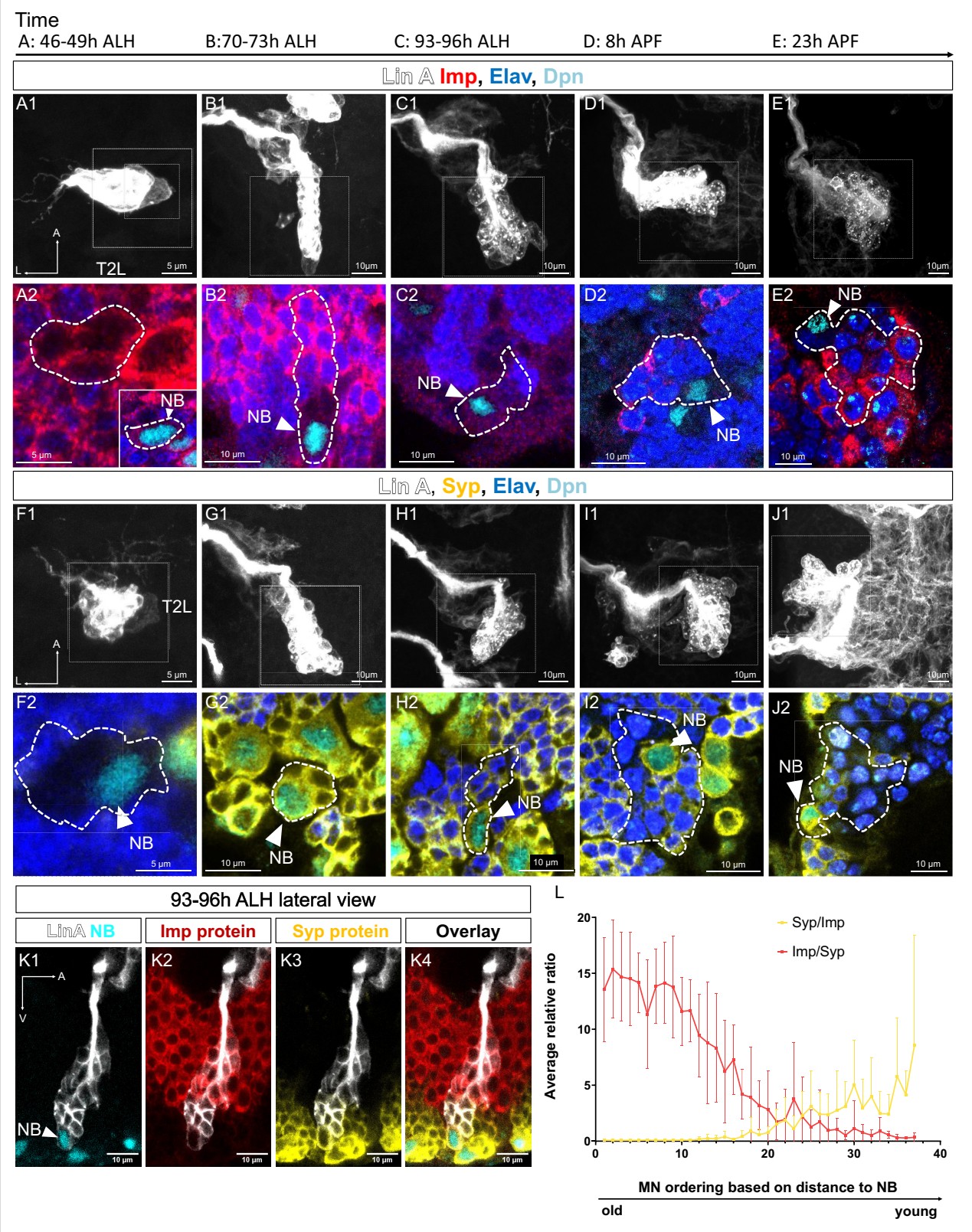

**Figure 7.** Spatiotemporal expression of Imp/Syp in Lin A/15. (**A1–J2**) Confocal images showing the development of Lin A/15 during larval and pupal stages, the developmental time points are indicated on top. ALH: after larva hatching; APF: after pupa formation. (**A1, B1, C1, D1, E1, F1, G1, H1, I1, J1**) Maximum projection of confocal sections of the second left thoracic hemisegment (T2L) where Lin A/15 is genetically labeled with mCD8::GFP (white). (**A2, B2, C2, D2, E2, F2, G2, H2, I2, J2**) Magnified views of boxed regions in (**A1, B1, C1, D1, E1, F1, G1, H1, I1, J1**) showing Lin A/15

*Figure 7 continued on next page*

Figure 7 continued

neuroblast (NB) and newborn motoneurons (MNs) immunostained with anti-Elav (neuronal marker, blue), anti-Dpn (NB marker, cyan), and anti-Imp (red) (**A2, B2, C2, D2, E2**) or anti-Syp (yellow) (**F2 G2, H2, I2, J2**). (**A2**) Inset: magnified view of the smaller boxed region in (**A1**) showing Lin A/15 NB. The dashed lines indicate GFP-labeled cells including NB and newborn MNs. White arrowheads indicate Lin A/15 NB. (**K1–K4**) Confocal section from the lateral view of a GFP Lin A/15 (white) at 93–96 hr ALH, immunostained with anti-Dpn (NB marker, cyan), anti-Imp (red), and anti-Syp (yellow). (**L**) Graph of relative expression levels of Imp and Syp protein in Lin A/15 post-mitotic MNs. The x axis represents MN ordering based on the distance to the NB. Note: Young MNs are closer to the NB while older MNs are further to the NB. Syp/Imp and Imp/Syp refer to the ratios of staining intensity values measured in ImageJ (N = 9). Error bars represent standard deviations.

The online version of this article includes the following source data for figure 7:

**Source data 1.** (7L) Immunostaining quantification: Imp and Syp staining signals in post-mitotic neurons.

expressed Imp in postmitotic neurons, including the supernumerary MNs, without affecting its expression in Lin A/15 NB, using the MARCM technique with the *VGlut-Gal4* enhancer trap driver. Under this experimental condition, more Lin A MNs were maintained in adult flies (***Figure 8L1–N***), implying a cell-autonomous function of Imp in promoting cell survival of MNs. Additionally, we generated Syp overexpression MARCM clones, and interestingly, we did not observe any significant effect on the number of MNs produced in adult flies (***Figure 8—figure supplement 1***). As a result, we decided not to delve further into Syp function in this particular genetic background.

In conclusion, our findings demonstrate that the opposite expression pattern of Imp and probably Syp in postmitotic neurons precisely shapes the size of Lin A/15 lineage by controlling the pattern of PCD in iMNs (***Figure 8***).

## The last-born MNs that are eliminated by PCD are primed with a specific combination of TFs under the control of Imp and Syp

In our previous study, we demonstrated that the first 29 MNs express a specific set of mTFs that determine the target muscle of each MN. Furthermore, we revealed that at least 5 out of the 16 TFs expressed in these first 29 MNs are post-transcriptionally regulated by the opposite gradients of Imp and Syp. Based on these findings, we hypothesized that the precise elimination of MNs could also result from a post-transcriptional regulation of TFs by Imp and Syp.

To investigate this, we analyzed the expression of Nvy and RunxA, two TFs known to be expressed in last-born surviving MNs, and Jim, a TF expressed in younger MNs in LL3, just before the elimination of the last-born MNs (***Guan et al., 2022c***). To achieve this, we utilized a new version of our computational tool (PCCD V2.0), which allows for a more precise determination of the TF code in last-born MNs (see material and methods). Our results revealed that last-born MNs express a specific combination of TFs, with high levels of Nvy and RunxA and no expression of Jim (***Figure 9A, E1, M***). Next, we manipulated the expression levels of Imp and Syp in Lin A/15.

By overexpressing Imp or knocking down Syp in Lin A/15 NB and its progeny, the number of Jim$^+$ MNs increased from 8 to 15 and 14 MNs, respectively (***Figure 9A1–A3, B1–B3, C1–C3***). Positive Cell Cluster Detection (PCCD) analysis demonstrated that last-born MNs express Jim de novo when Imp is overexpressed or Syp is knocked down (***Figure 9A4, B4, C4, M***).

RunxA is expressed in two cluster of MNs in young and in last-born MNs with high level of expression in the MNs eliminated by PCD during the pupal Stages (***Figure 9E1–E4, M***). The overexpression of Imp or the knowing down of Syp reduce drastically the number RunxA+ MNs (***Figure 9F1–F3, G1–G3, H1–H2***). The PCCD method reveals that the expression is completely abolished in last-born MNs while it is not affected in the youngest cluster of MNs when Imp is overexpressed or Syp knocked down (***Figure 9F4, G4, M***).

Nvy shows a similar expression pattern compared to RunxA (***Figure 9I1–I4, M***). The overexpression of Imp or the knockdown of Syp slightly reduces the number of Nvy+ MNs from 16 to 11 MNs (***Figure 9J1–L2***). PCCD analysis shows that the number of MNs expressing Nvy is only reduced in the last-born cluster of MNs when Imp is overexpressed or Syp is knocked down, while it is not affected in the youngest cluster of MNs (***Figure 9J4, K4, M***). Importantly, even though the expression of Nvy is not completely abolished in last-born MNs like RunxA, its expression is drastically reduced compared to control Lin A/15 (***Figure 9I2, J2, K2***).

Overall, these results demonstrate that overexpressing Imp or knocking down Syp changes the combination of TFs in last-born MNs from Nvy$^{high}$, RunxA$^{high}$, and Jim$^-$ to Nvy$^{low}$, RunxA$^-$, and Jim$^+$.

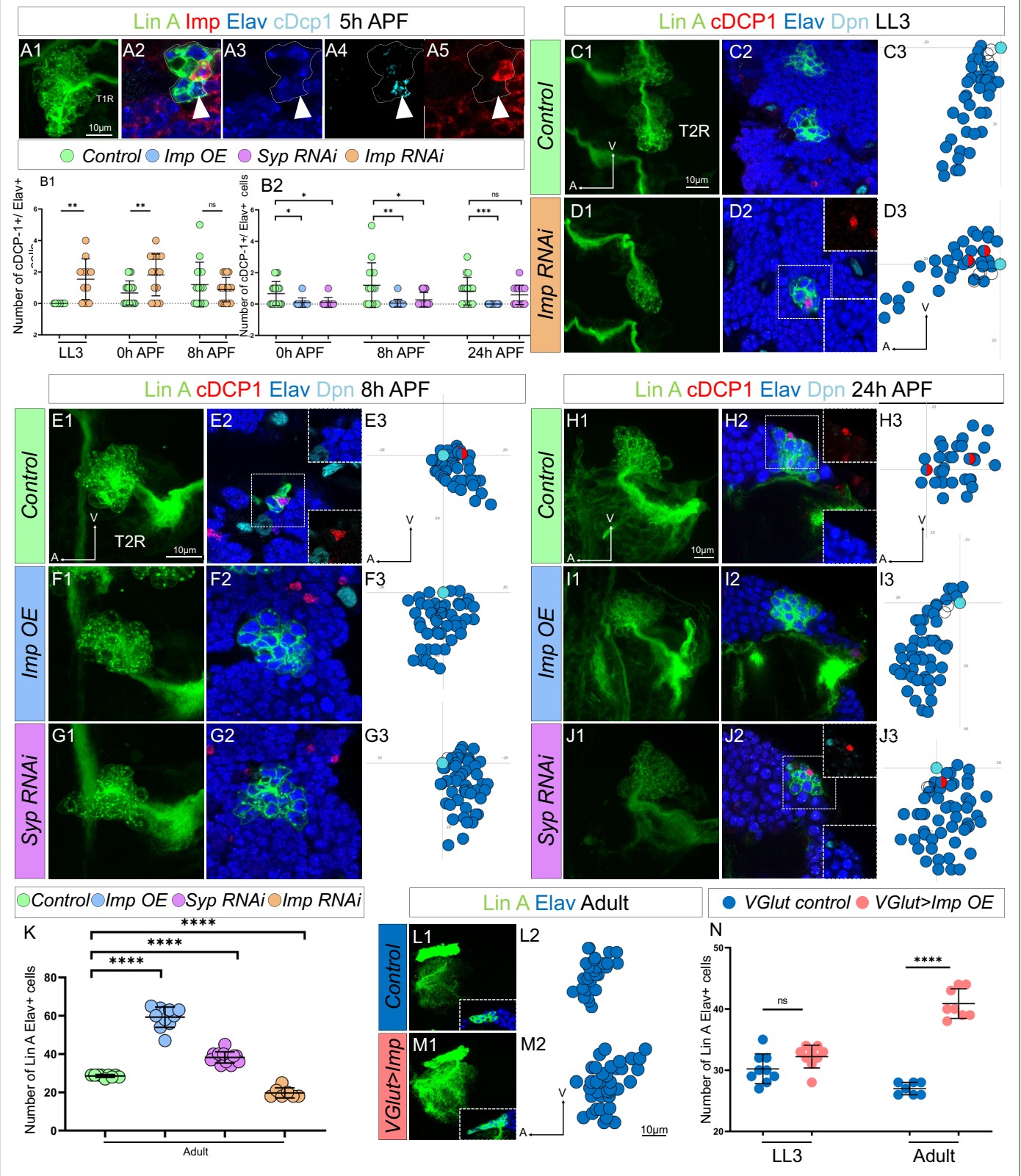

**Figure 8.** The opposite expression pattern in immature neurons of Imp/Syp instructs the number of surviving motoneurons (MNs). (**A1**) Maximum projection of confocal sections of the T1R hemisegment where Lin A/15 is genetically labeled with mCD8::GFP (green) at 5 hr APF. (**A2, A3, A4, A5**) are confocal sections of the Lin A/15 in (**A1**) showing the apoptotic MN immunostained with anti-Imp (red), anti-Elav (blue), and anti-cDcp1 (cyan) of (**A1**). Arrowheads indicate the apoptotic MN (Elav+, cDcp1+) is absent of Imp (Imp−). (**B1, B2**) Graph of the number of apoptotic MNs (Elav+, cDcp1+)

*Figure 8 continued on next page*

*Figure 8 continued*

observed in Lin A/15 at different developmental time points under different genetic conditions: *Control* (green), *Imp RNAi* (orange), *Imp OE* (blue), and *Syp RNAi* (purple). (**C1, D1, E1, F1, G1, H1, I1, J1**) Maximum projection of confocal sections of the second right thoracic hemi-segments (T2R) where Lin A/15 is genetically labeled with mCD8::GFP (green) under different genetic conditions *control* (**C1, E1, H1**), *Imp RNAi* (**D1**), *Imp OE* (F1, I1), and *Syp RNAi* (**G1, J1**). (**C2, D2, E2, F2, G2, H2, I2, J2**) Confocal sections of the second right thoracic hemisegment (T2R) in (**C1, D1, E1, F1, G1, H1, I1, J1**) immunostained with anti-cDcp1(red), anti-Elav neuronal marker (blue), and anti-Dpn (neuroblast [NB] marker, cyan). The boxed region in (**D2, E2, H2, J2**) indicated the presence of Elav+ cDcp1+ cells. (**C3, D3, E3, F3, G3, H3, I3, J3**) Graphs of the relative position of each Lin A/15 cell in (**C1, D1, E1, F1, G1, H1, I1, J1**) from a lateral perspective. Axes: Anterior (A), Ventral (V). Lin A/15 immature MNs are in blue, Lin A/15 ganglion mother cells (GMCs) are in white, Lin A/15 NB is in cyan, and cDcp1+ Elav+ neurons are in red and blue. (**K**) Graph of the number of Elav+ Lin A/15 neurons observed in adult flies under different genetic conditions: *control* (green), *Imp RNAi* (orange), *Imp OE* (blue), and *Syp RNAi* (purple). n ≥ 7. (**L1, M1**) Maximum projection of confocal sections of the left prothoracic neuromere (T1L) containing a control (**L1**) or a *VGlut >Imp* (**M1**) Lin A/15 MARCM clone. The boxed regions in (**L1–M1**) are confocal sections showing the Lin A/15 Elav+ (anti-Elav, Blue) GFP+ cells. (**L2, M2**) Graphs of the relative position of each Lin A/15 cell in (**L1, M1**) from a lateral perspective. Axes: Anterior (A), Ventral (V). Lin A/15 immature MNs are in blue. (**N**) Graph of the number of Elav+ *VGlut*+ Lin A/15 cells of control and *VGlut>Imp* Lin A/15 MARCM clones in third instar larvae (LL3) and adult flies. n ≥ 7. (**B, K, N**) Error bars represent standard deviations. Student's t test was performed to compare the difference between indicated groups. ns, P > 0.05, considered not significant; *, P ≤ 0.05; **, P ≤ 0.01; ***, P ≤ 0.001; ****, P ≤ 0.0001.

The online version of this article includes the following figure supplement(s) for figure 8:

**Figure supplement 1.** Syp overexpression (OE) does not change the number of motoneurons (MNs) produced by Lin A.

**Figure supplement 2.** No mutual inhibition between Imp and Syp in Lin A/15.

## Changing the combination of TF in post-mitotic MNs leads to survival of last-born MNs

Overexpressing Imp or knocking down Syp leads to the expression of a TF code in last-born MNs resembling the code found in earlier MNs (MNs 16–23), characterized by Nvy⁻, RunxA⁻, and Jim⁺. These findings suggest that the two RBPs, Imp and Syp, may control the number of surviving MNs through TF regulation.

To further investigate this possibility, we used the MARCM technique to modify the TF code of last-born MNs without affecting the expression of Imp and Syp. In our genetic background, control MARCM clones typically produce around 26 MNs instead of 28 (*Figure 10A1–A2, E*). We observed that in different genetic backgrounds, Lin A/15 sometimes produces fewer than 28 MNs.

When we overexpressed Jim (*Figure 10C1–C3*), including in the last-born MNs, or removed Nvy (*Figure 10B1–B3*), the number of MNs produced by Lin A/15 increased to approximately 32 (*Figure 10E*). Notably, in nvy−/− Lin A/15 MARCM clones overexpressing Jim (*Figure 10D1–D3*), the number of MNs produced is close to 50 (*Figure 10E*).

These results demonstrate that imposing a TF code in last-born MNs resembling the code found in early-born MNs enables their survival until the adult stage. This suggests that the specific combination of TFs controlled by Imp and Syp plays a crucial role in determining the fate and survival of motor neurons during development.

## Discussion
### Imp and Syp, lineage size ruler

The timing of neurogenesis termination plays a crucial role in determining the number of neurons produced by stem cells during development. Using our genetic tool to trace a single lineage, we demonstrated that altering the timing of neural stem cell decommissioning changes the number of neurons generated from that stem cell. We found that two intrinsic temporal factors, Imp and Syp, actively participate in defining the final clonal size of a lineage by signaling the timely apoptosis of NBs. Previous research has indicated that both intrinsic mechanisms and external cues control the expression pattern of Imp and Syp in brain NBs. In some brain lineages, Imp and Syp cross-repress each other (*Liu et al., 2015*; *Syed et al., 2017*), and this cross-repression is influenced by external cues such as the ecdysone hormone (*Homem et al., 2014*) or activins (*Rossi and Desplan, 2020*). However, in Lin A/15 NB, we observed that there is no mutual inhibition between Imp and Syp, suggesting that Imp/Syp-independent mechanisms regulate their sequential expression in Lin A/15 NB (*Figure 8—figure supplement 2*).

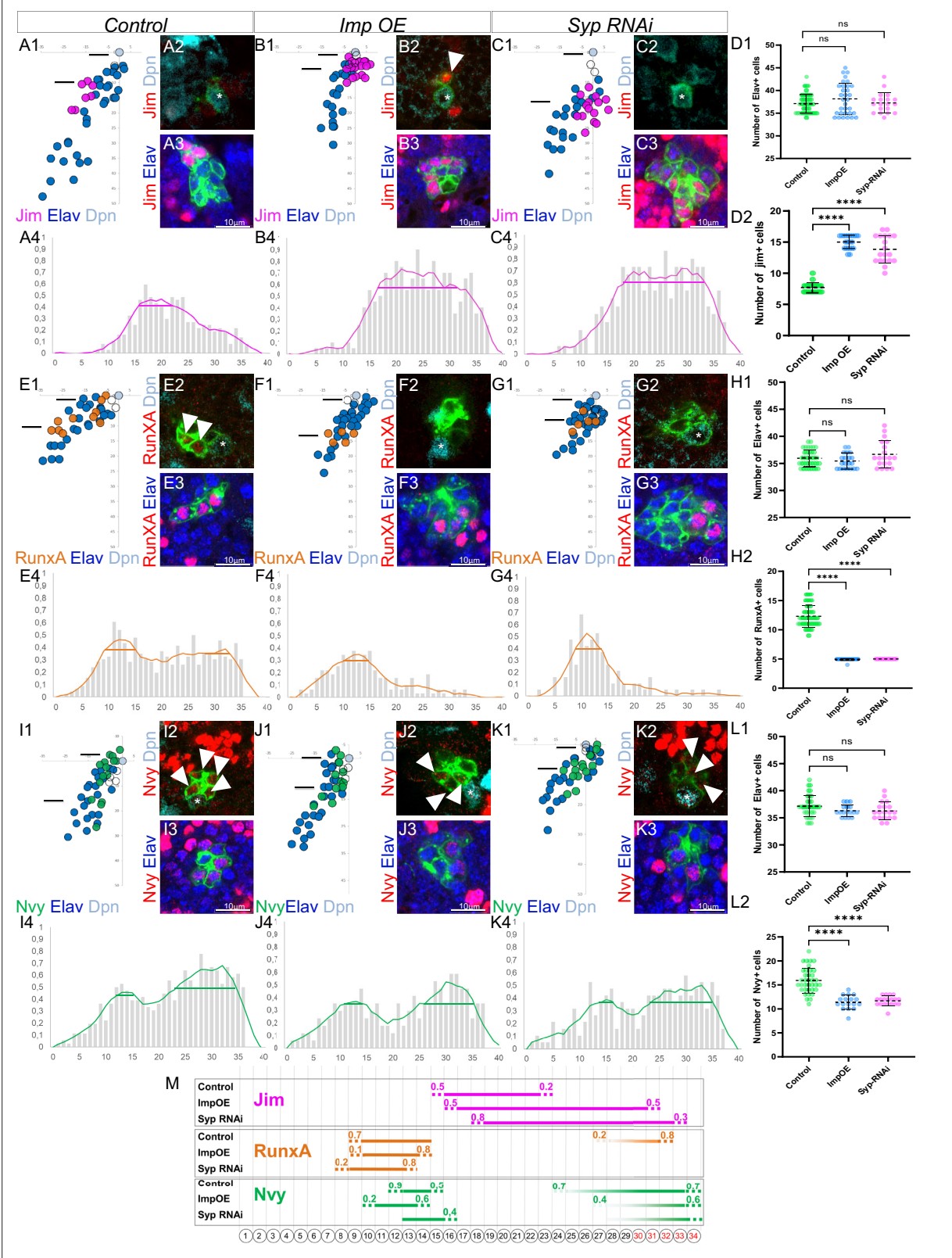

**Figure 9.** The last-born motoneurons (MNs) eliminated by PCD are primed with a specific combination of transcription factors (TFs) under control of Imp and Syp. (**A1, B1, C1, E1, F1, G1, I1, J1, K1**) Plots of the relative position of each Lin A/15 cell from a lateral perspectives showing the expression of Jim (**A1, B1, C1**), RunxA (**E1, F1, G1**), and Nvy (**I1, J1, K1**) in purple, orange, and green, respectively, the Elav+ MNs (blue), the neuroblast (NB) (cyan), and the ganglion mother cell (GMC) in control (**A1, E1, I1**), Imp overexpression (OE) (**B1, F1, J1**), and Syp RNAi (**C1, G1, K1**). (**A2–A3, B2–B3,**

*Figure 9 continued on next page*

*Figure 9 continued*

C2–C3, E2–E3, F2–F3, G2–G3, I2–I3, J2–J3, K2–K3) confocal sections showing the expression of Jim (**A2–A3, B2–B3, C2–C3**), RunxA (**E2–E3, F2–F3, G2–G3**), and Nvy (**I2–I3, J2–J3, K2–K3**) in red, the Elav+ MNs (blue), the Dpn+ NB (cyan). The position of the sections are shown in (**A1, B1, C1, E1, F1, G1, I1, J1, K1**). The asterisk indicate the NBs. Note 1: The arrowhead in B2 indicates a jim+ neuron close to NB, this is never seen in the control Lin A/15. Note 2: The arrowheads in E2 indicate RunxA+ neurons close to NB, this is never seen in Imp OE and Syp RNAi Lin A/15. Note 3: The arrowheads in I2, J2, and K2 indicate NVy+ neurons close to the NB, the expression of Nvy is barely detectable in Imp OE and Syp RNAi Lin A/15. (**A4, B4, C4, E4, F4, G4, I4, J4, K4**) Graphs of the frequency of Jim (**A4, B4, C4**), RunxA (**E4, F4, G4**) and Nvy (**I4, J4, K4**) expression as a function of *x' x'*: MN ordering axis according to their relative distance from the NB among >15 specimens before (gray bare) and after (colored lines) applying a Savitzky–Golay filter (see Material and methods) in control (**A4, E4, I4**), Imp OE (**B4, F4, J4**), and Syp RNAi (**C4, G1, K4**). The horizontal bar indicates the Jim⁺ cell cluster detected with the Positive Cell Cluster Detection (PCCD) method. (**D1–D2, H1–H2, L1–L2**) Graphs of the number of Elav+ Lin A/15 MNs (**D1, H1, L1**) and graphs of the number of Elav+ Lin A/15 MNs expressing Jim (**D2**), RunxA (**H2**), and Nvy (**L2**) in control, Imp OE, and Syp RNAi Lin A/15. n ≥ 17. Error bars represent standard deviations. Student's t test was performed to compare the difference between indicated groups. ns, P > 0.05, considered not significant; ****, P ≤ 0.0001. (**M**) Schematic of the TF codes expressed in each immature MN (iMN) predicted by the PCCD method in an L3 larva in control, Imp OE, and Syp RNAi Lin A/15. Bottom: Schematic of the cell body of Lin A/15 iMNs. The numbers inside indicate their relative distances from NB. Top: The horizontal bars indicate the TF+ cell clusters detected with the PCCD method. The dotted lines indicate the coverage index at the border (see Materials and methods).

Furthermore, our investigation of a single neuronal lineage revealed that the opposite expression pattern of Imp and Syp in postmitotic neurons is also important in determining lineage size by regulating PCD. The opposite expression levels of Imp and Syp in iMNs correlate with their temporal expression in the NB: first-born neurons express high levels of Imp, while last-born neurons express high levels of Syp. Interestingly, both Imp and Syp are actively expressed in iMNs, indicating that their expression in these neurons is not simply a consequence of their expression in the NB (*Guan et al., 2022c*).

We question whether iMNs maintain their Imp/Syp expression by inheriting determinants from the NB. If so, are these determinants directly influencing Imp/Syp expression, or do they give MNs the capacity to respond to external cues, such as the ecdysone hormone? Based on our hypothesis, we propose that postmitotic MNs, similar to their state in the NB, retain the ability to respond to external cues, which could explain why MNs are eliminated during early pupal stages when ecdysone is highly expressed (*Warren et al., 2006*).

## Imp and Syp, highly versatile RBPs in specifying neuronal identity and lineage sizes

Imp and Syp are versatile proteins that play roles in various aspects of NB development. Their opposite temporal expression in brain NBs controls the temporal identity of the NB (*Liu et al., 2015*; *Syed et al., 2017*), the timing of NB decommissioning (*Yang et al., 2017*), and the speed of cell division (*Samuels et al., 2020*). In the brain, their opposing temporal expression in mushroom body NBs shapes the expression pattern of the Chinmo TF, which, in turn, determines the identity of the neuronal progeny based on its concentration.

Additionally, Imp and Syp appear to regulate the expression of terminal selector genes. Originally defined in *C. elegans*, terminal selectors are TFs that maintain the expression of proteins crucial for neuron function, such as neurotransmitters or neuropeptides (*Allan et al., 2005*; *Eade et al., 2012*; *Hobert, 2011*; *Hobert, 2016*). In the mushroom body, Imp and Syp shape the terminal molecular features by regulating the terminal selector Mamo (*Liu et al., 2019*).

We have recently revealed that Imp and Syp not only control the morphology of Lin A/15 MNs by determining the temporal identity of the NB but also by shaping a combination of mTFs in postmitotic MNs, which subsequently control their morphologies (*Guan et al., 2022c*). Here, we show that MNs eliminated by PCD during metamorphosis express a different combination of TFs compared to surviving MNs. Furthermore, we demonstrate that changing the opposite expression of Imp and Syp in Lin A/15 alters the TF code not only in the surviving MNs but also in the MNs eliminated by apoptosis. These MNs adopt a TF code similar to that of MNs that survive, suggesting that the regulation of PCD in iMNs by Imp and Syp is dependent on TFs. To test this hypothesis, we changed the TF code without affecting Imp/Syp expression and were able to prevent the elimination of the last-born MNs. Taken together, all these data highlight that both RBPs control two parameters in postmitotic neurons: neuronal diversity and neuronal survival. These two parameters are directly coordinated through the regulation of TFs by Imp and Syp.

The question of why an excess of MNs is produced and why neurons undergo PCD during development is a fundamental one in neurodevelopment. PCD is observed in various animal models studied in laboratories, from *C. elegans* to vertebrates, and several explanations have been proposed to understand why such mechanisms have been selected during evolution (*Dekkers et al., 2013*; *Oppenheim, 1991*). The widespread occurrence of PCD during neurodevelopment leads to an intriguing hypothesis that neurons normally fated to die represent an important reservoir that can be used during evolution to explore different morphological possibilities. In this scenario, different mTF codes can be tested without affecting the axon–muscle connectome, allowing for the exploration of various combinations of mTFs until a functional combination is selected. This process may contribute to the diversification of neuronal morphologies and functions, ultimately shaping the complexity of the nervous system during evolution.

### Imp/Syp proteins in development and disease

Imp and Syp are evolutionarily conserved, both homologs are highly expressed in the developing mouse brain and play vital roles in neural development, suggesting a fundamental conservation of their function in the development of central nervous (*Chen et al., 2012*; *Mori et al., 2001*; *Williams et al., 2016*). For example, Imp1, one of the three mouse orthologs of Imp family, is highly expressed in young neuronal progenitors. Its temporal expression with other RBP partners changes the temporal identity of the neuronal stem cells. In particular Imp promotes the self-renewal state of neuronal stem cells while inhibiting differentiation genes (*Nishino et al., 2013*).

As mentioned in the introduction, any dysregulation of the machinery controlling neuronal stem cell physiology can have dramatic consequences. For example dysregulation of the RBPs expression are a common feature of neurodegenerative diseases (*Lenzken et al., 2014*). Interestingly, the Imp family also plays a key role in the stem cell physiology of many other organs and several studies have revealed that the Imp family maintain the proliferative state of different type of cancers (*Degrauwe et al., 2016*; *Bell et al., 2013*; *Genovese et al., 2019*; *Lan et al., 2021*; *Sun et al., 2021*). The powerful genetic tools available in *Drosophila* allow us to decipher their functions in the stem cells versus postmitotic neurons, which is a first step not only to better understand how the organ is built, but also to decipher the basis of brain carcinogenesis and neurodegenerative diseases.

## Materials and methods

**Key resources table**

| Reagent type (species) or resource | Designation | Source or reference | Identifiers | Additional information |
|---|---|---|---|---|
| Genetic reagent (*D. melanogaster*) | UAD-KD (attP2[68A4]) | *Awasaki et al., 2014* | N/A | |
| Genetic reagent (*D. melanogaster*) | Dpn>KDRT-stop-KDRT>CRE (su(Hw) attP8[8E10]) | *Awasaki et al., 2014* | N/A | |
| Genetic reagent (*D. melanogaster*) | act>loxP-stop-loxP>LexA::P65 (attP40[25C7]) | *Lacin and Truman, 2016* | N/A | |
| Genetic reagent (*D. melanogaster*) | lexAop-myr::GFP (su(Hw)attP5[50F1]) | *Awasaki et al., 2014* | N/A | |
| Genetic reagent (*D. melanogaster*) | R10c12-GAL4 (3rd chromosome, attp2) | *Lacin and Truman, 2016* | N/A | |
| Genetic reagent (*D. melanogaster*) | lexAop-Imp-RNAi (attP40[25C7]) | *Rueden et al., 2017* | N/A | |
| Genetic reagent (*D. melanogaster*) | lexAop-Imp-RM (attP40[25C7]) | *Rueden et al., 2017* | N/A | |
| Genetic reagent (*D. melanogaster*) | lexAop-Syp-RNAi (attP40[25C7]) | *Rueden et al., 2017* | N/A | |
| Genetic reagent (*D. melanogaster*) | tub-gal4 (3rd chromosome,79A2) | Bloomington *Drosophila* Stock Center | BDSC: 5138 | |
| Genetic reagent (*D. melanogaster*) | DVGlut-Gal4 (2nd chromosome, 22E1) | Bloomington *Drosophila* Stock Center | BDSC: 26160 | |

*Continued on next page*

*Continued*

| Reagent type (species) or resource | Designation | Source or reference | Identifiers | Additional information |
|---|---|---|---|---|
| Genetic reagent (*D. melanogaster*) | DVGlut-LexA::GAD (VGlutMI04979) | This study | N/A | |
| Genetic reagent (*D. melanogaster*) | UAS-P35 (2nd and 3rd chromosome) | Bloomington *Drosophila* Stock Center | BDSC: 5072; 5073 | |
| Genetic reagent (*D. melanogaster*) | 20XUAS-Imp-RM-Flag (3rd chromosome) | *Liu et al., 2015* | N/A | |
| Genetic reagent (*D. melanogaster*) | actin^FRT-stop-FRT^Gal4 | Gift from Alain Garces | N/A | |
| Antibody | anti-Dcp1 (rabbit polyclonal) | Cell signaling | Cat#9578S; RRID:AB_2721060 | 1:50 |
| Antibody | anti-Elav (mouse monoclonal) | DSHB | Cat#9F8A9; RRID:AB_2314364 | 1:50 |
| Antibody | anti-Elav (rat monoclonal) | DSHB | Cat#7E8A10; RRID:AB_528218 | 1:50 |
| Antibody | anti-repo (mouse monoclonal) | DSHB | RRID:AB_528448 | 1:100 |
| Antibody | anti-PH3 (rabbit polyclonal) | Abcam | RRID:AB_2164915 | 1:50 |
| Antibody | anti-Dpn (guinea-pig, unknown clonality) | Gift from Jim Skeath | RRID:AB_2314299 | 1:250 |
| Antibody | anti-Imp (rat, unknown clonality) | Gift from Claude Desplan | N/A | 1:200 |
| Antibody | anti-Syp (rabbit, unknown clonality) | Gift from Claude Desplan | N/A | 1:400 |
| Antibody | anti-mouse Alexa 647 (goat, polyclonal) | Invitrogen | Cat#A32728; RRID:AB_2633277 | 1:250 |
| Antibody | anti-rat Alexa 647 (donkey, polyclonal) | Jackson | Cat#712-605-153; RRID:AB_2340694 | 1:250 |
| Antibody | anti-mouse Alexa 555 (goat, polyclonal) | Invitrogen | Cat#A32727; RRID:AB_2633276 | 1:250 |
| Antibody | anti-rabbit Alexa 555 (goat, polyclonal) | Invitrogen | Cat#A32732; RRID:AB_2633281 | 1:250 |
| Antibody | anti-Rat Alexa 555 (goat, polyclonal) | Abcam | Cat#ab150166 | 1:250 |
| Antibody | anti-guinea-pig DyLight405 (donkey, polyclonal) | Jackson | Cat#706-475-148; RRID:AB_2340470 | 1:250 |
| Recombinant DNA reagent | pBS-KS-attB2-SA(2)-T2A-LexA::GADfluw-Hsp70 vector | Addgene | Cat#78307 | |
| Chemical compound, drug | Formaldehyde | Thermo Fisher | Cat#28908 | |
| Chemical compound, drug | PBS | Dutscher | Cat#X0515-500 | |
| Chemical compound, drug | Triton X-100 | Sigma | Cat#T8787-100mL | |
| Chemical compound, drug | Tween 20 | Sigma | Cat#P7949-100ML | |
| Chemical compound, drug | BSA | Sigma | Cat#A7906-500 g | |
| Chemical compound, drug | Vectashield mounting medium | Vector Laboratories | Cat#H1000 | |
| Chemical compound, drug | Abberior MOUNT SOLID ANTIFADE | abberior GmbH | MM-2013-2X15ML | |
| Chemical compound, drug | Salmon sperm DNA | Thermo Fisher | Cat#15632011 | |
| Chemical compound, drug | Yeast tRNA | Thermo Fisher | Cat#AM7119 | |
| Chemical compound, drug | Murine RNase inhibitor | New England Biolabs | M0314L | |
| Software, algorithm | Amira 3D software (version 6.2) | SCR_007353 | https://www.fei.com/ | |
| Software, algorithm | ImageJ (version 1.48) | *Schneider et al., 2012* | https://imagej.nih.gov/ij/ | |
| Software, algorithm | GraphPad (Prism 8) | GraphPad Software Inc | https://www.graphpad.com/scientific-software/prism/ | |
| Software, algorithm | Matlab (2021b) | MathWorks | https://www.mathworks.com/ | |
| Software, algorithm | smFISH analysis source code in Matlab | *Guan et al., 2022c* | https://github.com/Wenyue2022/smFish-analysis-source-code; *Guan, 2022a* | |
| Software, algorithm | PCCD note book | *Guan et al., 2022c* | https://github.com/Wenyue2022/PCCD-note-book; *Guan, 2022b* | |

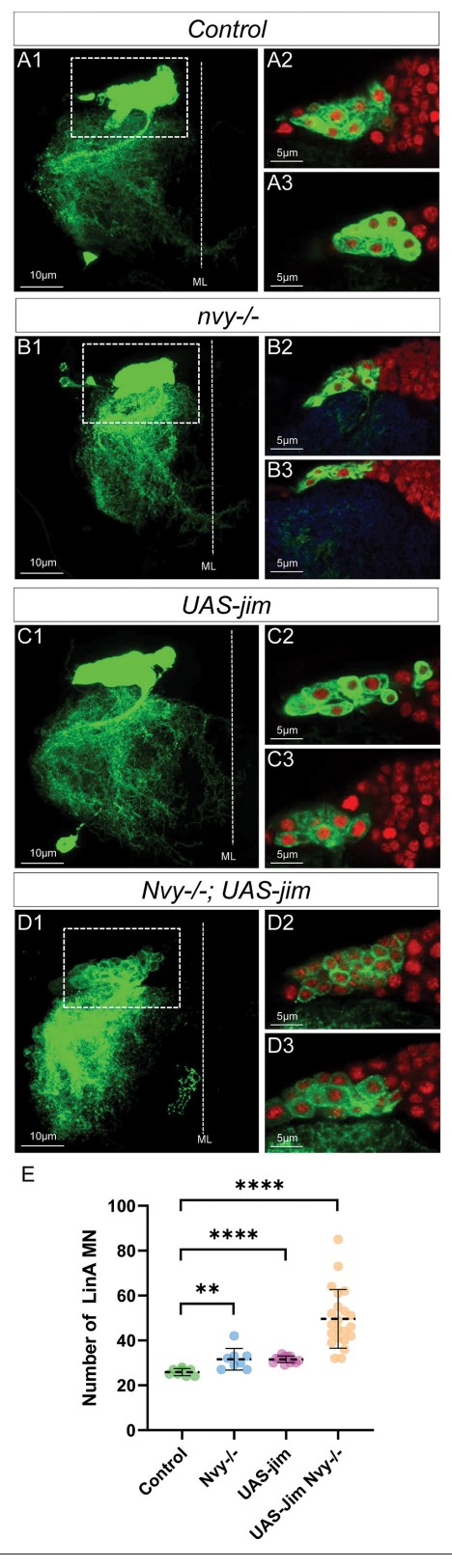

**Figure 10.** Changing the combination of transcription factor (TF) in last-born motoneurons (MNs) leads to MN survival. (**A1–D3**) Maximum projection of confocal sections of a right prothoracic hemisegment (T2R)

*Figure 10 continued on next page*

*Figure 10 continued*

with a Lin A/15 MARCM clone genetically labeled with mCD8::GFP (green) under the control of *VGlut-Gal4* (**A1, B1, C1, D1**) and confocal sections trough Lin A/15 (green) and labeled with Elav (red) (**A2–A3, B2–B3, C2–C3, D2–D3**) in different genetic conditions: control (**A2–A3**), *nvy−/−* (**B2–B3**), *UAS-jim* (**C2–C3**), and *nvy−/−; UAS-jim* (**D2–D3**). (**E**) Graphs of the number of Elav+ VGlut+ MNs in control, *nvy−/−* (**B2–B3**), *UAS-Jim* (**C2–C3**), *nvy−/−; UAS-jim* (**D2–D3**) MARCM clones. n ≥ 8. Error bars represent standard deviations. Student's t test was performed to compare the difference between indicated groups. \*\*, P ≤ 0.01; \*\*\*\*, P ≤ 0.0001.

## Fly husbandry and crosses

Unless otherwise described, flies were maintained on a standard cornmeal medium and kept at 25°C in a 12:12-hr light:dark cycle. Males and females were chosen at random. A full list of strains used in the paper is included in the Key resources table.

## Lin A/15 NB tracing system (all figures)

Lin A/15 restrictive labeling is achieved by immortalizing Gal4 expression in Lin A/15 NBs and its descendants (*Awasaki et al., 2014*; *Lacin and Truman, 2016*). The following fly strains were crossed to specifically label Lin A:10c12-GAL4 crossed with *Dpn>KDRT-stop-KDRT>CRE; act>loxP-stop-loxP>LexA::P65; lexAop-myr::GFP; UAS-KD*.

The following fly strains were used to overexpress and knockdown genes (*Figures 5 and 8*):

*Dpn >KDRT-stop-KDRT>CRE; act>loxP-stop-loxP>LexA::P65; lexAop-myr::GFP; UAS-KD* crossedwith *LexAop-Imp-Flag/CyO; R10c12-GAL4/TM6B* to overexpress Imp;with *LexAop-Syp-RNAi/CyO; R10c12-GAL4/TM6B* to knockdown Syp;with *LexAop-Imp-RNAi/CyO; R10c12-GAL4/TM6B* to knockdown Imp.

## Lin A/15 MARCM

Genetic crosses to label with GFP all Lin A cells and with mCherry VGlut+ Elav+ neurons (*Figure 2*):

*y,w, hs-Flp1.22; VGlut-LexA::GAD, FRT42D, LexAop-mCherry/CyO; tub-GAL4/TM6B* crossed to *y,w, hs-Flp1.22; VGlut-Gal4, UAS-mCD8::GFP, Mhc-RFP, FRT42D, tub-Gal80/CyO; UAS-mCD8::GFP/MKRS.*

Genetic crosses to inhibit apoptosis in immature postmitotic neurons (*Figure 6*):

*y,w,hs-Flp[1.22]; VGlut-Gal4, UAS-mCD8::GFP, Mhc-RFP FRT42D/CyO; TM6B/MKRS* crossed to *y,w,hs-Flp[1.22]; VGlut-GAL4, UAS-mCD8::GFP, Mhc-RFP, FRT42D, tub-Gal80/CyO; UAS-P35/TM6B.*

Genetic crosses to overexpress Imp only in postmitotic neurons (**Figure 8**):

*y,w,hs-Flp[1.22]; VGlut-Gal4, UAS-mCD8::GFP, Mhc-RFP, FRT42D/CyO; TM6B/MKRS* crossed to *y,w,hs-Flp[1.22]; VGlut-Gal4, UAS-mCD8::GFP, Mhc-RFP, FRT42D, tub-Gal80/CyO; UAS-Flag-Imp-RM/ TM6B.*

Genetic crosses to overexpress Syp only in postmitotic neurons (**Figure 8—figure supplement 1**):

*y,w, hs-Flp1.22; VGlut-LexA::GAD, FRT42D, LexO-mCherry/CyO; tub-GAL4/TM6B* crossed to *y,w, hs-Flp1.22; FRT42D, tub-gal80/CyO; 20XUAS-Syp-RB-HA/TM6B.Tb.*

Genetic crosses to generate *nvy−/−* and *nvy−/−* Jim overexpression Lin A clones (**Figure 10**):

*y, w, hs-Flp, UAS-mCD8::GFP; VGlut-Gal4, FRT42D, nvy−/CyO; 10XUAS-myr::GFP/MKRS* crossed to *y,w, hs-Flp1.22; VGlut-Gal4, UAS-mCD8::GFP, Mhc-RFP, FRT42D, tub-Gal80/CyO ; UAS-Jim CDS/ TM6b.*

Genetic crosses to generate Jim overexpression and Control Lin A clones (**Figure 10**):

*y,w, hs-Flp1.22; VGlut-Gal4, UAS-mCD8::GFP, Mhc-RFP, FRT42D/CyO; TM6B/MKRS* Crossed to *y,w, hs-Flp1.22; VGlut-Gal4, UAS-mCD8::GFP, Mhc-RFP, FRT42D, tub-Gal80/CyO; UAS-jim CDS/TM6B.*

First-instar larvae (0–12 hr ALH) were heat shocked at 37°C for 20 min to induce mosaic clones in L3 larvae and at 35°C for 15 min to induce mosaic clones in adults.

### *VGlut-LexA::GAD* transgenic line

The *VGlut-LexA::GAD* transgenic line is generated by the Trojan-mediated conversion of MIMIC (Trojan-MiMIC) technique (**Diao et al., 2015**). A *pBS-KS-attB2-SA(2)-T2A-LexA::GADfluw-Hsp70* Plasmid (addgene plasmid #78307) was injected into embryos of flies bearing intronic MiMIC inserts at *VGlut* gene (VGlutMI04979) together with phiC31 integrase on the genetic background. G0 flies were crossed with flies from the *y, w; Sp/CyO; Dr/TM3, Sb* double balancer line, and y- recombinant progeny, which had lost the y+ selection marker associated with the MiMIC insertion, were isolated. The integrase-dependent exchange of *T2A-LexA::GAD*-containing cassette produce a *LexA::GAD* driver line that having an expression pattern corresponding to that of *VGlut*.

### Immunohistochemistry

#### Immunostaining of larval and pupal CNS

Inverted L3 larvae or open pupae were fixed in 4% paraformaldehyde in Phosphate-buffered saline (PBS) for 20 min at room temperature and blocked in the blocking buffer for 1 hr. L3 larval or pupal CNS were carefully dissected in PBS and then incubated with primary antibodies overnight (≥12 hr) and secondary antibodies in dark for 1 day (≥12 hr) at 4°C. Fresh PBST–BSA (PBS with 0.1% Triton X-100, 1% bovine serum albumin) was used for the blocking, incubation, and washing steps: five times for 20 min at room temperature after fixation and after primary/secondary antibodies. Larval/pupal CNS were mounted onto glass slides using Vectashield anti-fade mounting medium (Vector Labs). Slides were either imaged immediately or stored at 4°C.

#### Immunostaining of adult VNC

After removing the abdominal and head segments, thoraces of the flies were opened and fixed in 4% paraformaldehyde in PBS for 25 min at room temperature and blocked in the blocking buffer for 1 hr. After dissection, adult VNC were incubated with primary antibodies for 1 day and secondary antibodies in dark for 1 day at 4°C. Fresh PBST–BSA (PBS with 0.1% Triton X-100, 1% BSA) was used for the blocking, incubation, and washing steps: five times for 20 min at room temperature after fixation and after primary/secondary antibodies. VNC were mounted onto glass slides using Vectashield anti-fade mounting medium (Vector Labs). Slides were either imaged immediately or stored at 4°C.

#### Primary and secondary antibodies

The primary antibodies used in this study include: mouse anti-Elav (DSHB-9F8A9), rat anti-Elav (DSHB-7E8A10), mouse anti-Repo (DSHB- 8D12), rabbit anti-cDcp1 (CellSignaling-9578), rabbit anti-PH3 (Abcam-ab80612), guinea-pig anti-Dpn (gift from Jim skeath, **Skeath et al., 2017**), rat anti-Imp, and rabbit anti-Syp (gifts from Chris Doe).

The secondary antibodies used in this study include: goat anti-Mouse Alexa 647 (Invitrogen-A32728), donkey anti-rat Alexa 647 (Jackson- 712-605-153), goat anti-Mouse Alexa 555 (Invitrogen-A32727),

goat anti-Rabbit Alexa 555 (Invitrogen-A32732), goat anti-Rat Alexa 555 (Abcam-ab150166), and donkey anti-guinea-pig DyLight405 (Jackson- 706-475-148).

## Image acquisition

Multiple 0.5-μm-thick (with exceptions of 1-μm-thick for *Figure 6H–M*) sections in the *z* axis (ventro-dorsal for larval/pupal CNS or adult VNC) were imaged with a Leica TCS SP8 or a Zeiss LSM 780 confocal microscope. Binary images for z stack images were generated using NIH ImageJ.

## 5-ethynyl-2'deoxyuridine (EdU) labeling (Figure 6)

To mark late-born MNs, mid-third-instar larvae (98–104 hr ALH) were transferred from standard fly food to fly food containing 250 mM EdU. Pupae were then dissected at indicated time points. Open pupae were fixed in 4% paraformaldehyde in PBS for 20 min at room temperature, followed by a quick wash with PBST (PBS with 0.1% Triton X-100). Edu labeling was then detected using Clicl-iT EdU imaging kit (Invitrogen) according to the manufacturer's instructions. The immunostaining was then performed as described in the Immunostaining section.

## NB volume quantification

Each Lin A/15 NB was segmented in 3D in ImageJ/Fiji (*Rueden et al., 2017*; *Schindelin et al., 2012*; *Schneider et al., 2012*), using the LimeSeg plugin (*Machado et al., 2019*), on the GFP channel, with the following parameters: D_0~4–6, F pressure = 0.01, Z_scale = 6.8, Range in d0 units ~4–6, number of integration step = −1, real *XY* pixel size = 50. The volume of each segmented cell was used to make the graph in (*Figure 3—figure supplement 1*).

## Quantification and statistical analysis

Graphs of the relative position of each Lin A/15 cell were generated with Microsoft Excel. The spatial coordinates were assigned to each cell using the cell counter plug-in of NIH ImageJ software. The coordinates of each cell were normalized with Microsoft Excel in order to have the position of the Lin A NB at the origin of the plot graph. For samples without NB labeled (as in *Figures 6C1 and 7H3*), the coordinates of each cell were then normalized to a cell located ventrally (as the cDcp1 positive cell in *Figures 6C1 and 7H3*).

Graphs of quantification and comparison were generated with Prism (GraphPad Software). All bar errors represent standard deviation of a minimum of seven samples, each dot represents a single sample analyzed. Otherwise, the sample size used for each genotype is indicated on the graph or in the text and/or the figures legends. Student's *t*-test (*Figures 1F, 6D, F, G, and 8B, K, N*) or Fisher's test (*Figure 5P, Q*) was performed to compare the difference in between indicated groups. Differences of $p < 0.05$ were considered significant. *$0.01 < p < 0.05$; **$0.001 < p < 0.01$; ***$0.0001 < p < 0.001$; ****$p < 0.0001$.

The quantification process of relative expression of Imp and Syp in *Figure 7L* was described below: In ImageJ Plugin – Cell Counter, mark the Lin A NB. Draw polygonal Regions of interest (ROIs) of Lin A post-mitotic cells at their sagittal planes, save into ROI manager by clicking 'add'. This will record both the ROI and the z slice. Rename ROIs as pmn1, pmn2, pmn3, etc. While drawing the ROIs, use Cell Counter to register spatial coordinates of all Lin A cells, assign each cell to a type, measure and copy into Excel template. Calculate the magnitudes of $V = (x, y, z)$. Sort the spreadsheet by *V*, which is the distance to NB. For each ROI, measure mean intensity in each channel. Calculate protein level ratios in Excel and plot in Prism.

All schematics were made in Microsoft PowerPoint (*Figures 1G, 2H, 3G, and 6L*).

## smFISH
### Probe design and preparation for smFISH

smFISH probe design principle in this study was described in our previous work (*Guan et al., 2022c*). Briefly, primary probes against exonal sequences of *Imp*, *Syp* and *Dpn* (common sequences of all isoforms of genes of interest; up to 48 probes per gene) were designed using the Biosearch Technologies stellaris RNA FISH probe designer tool (free with registration, https://biosearchtech.com). The primary probe sequences for each gene used in this study are shown in *Supplementary file 1*. Three Flap sequences were used in this study: the X flap sequence (CACTGAGTCCAGCTCGAAAC

TTAGGAGG) used in *Tsanov et al., 2016*, and two sequences we designed and named: RS3505 ( AACTACATACTCCCTACCTC) and RS0406 (ACCCTTACTACTACATCATC). The reverse complements of these Flap sequences were added to the 5' end of the primary probe sequences, using one Flap sequence for all primary probes targeting one gene to avoid signal crosstalk. The primary probe sets were purchased from Integrated DNA Technologies (IDT), using 25 nmol synthesis scale, standard desalting, and at 100 µM in nuclease-free H2O. RS3505 conjugated with Atto565 and RS0406 conjugated with Atto700 are synthesized by LGC Biosearch Technologies and Bio-Synthesis Inc, respectively. X-Flap conjugated with Alexa 647 was a gift from Tom Pettini, University of Cambridge.

## smFISH, sample preparation, and hybridization

Dissected larval and pupal CNS from were fixed in 4% paraformaldehyde (in PBS with 0.3% Triton X-100) for 20 min at room temperature. Samples were washed for three times of at least 15 min with PBSTW (PBS with 0.3% Tween-20) before a pre-hybridization wash in smiFISH wash buffer (4 M urea in 2× Saline-sodium citrate buffer (SSC)) at 37°C for 30 min. Samples were then incubated with primary probe sets diluted in smiFISH hybridization buffer 4 M urea, 2× SSC, 5% (vol/vol) dextran sulfate, 0.2 mg/ml 1:1 mixture of sheared salmon sperm DNA and yeast tRNA, 1% (vol/vol) murine RNase inhibitor at 37°C for 12–14 hr. Samples were washed for 40 min at 37°C followed by three times of 15 min in smiFISH wash buffer at room temperature, then incubated in secondary (Flap) probe sets diluted in smiFISH hybridization buffer for 1 hr. Samples were then washed for 2 min in smiFISH wash buffer and three times of 10 min in 2× SSC, then mounted in Vectashield or Abberior Solid mounting media.

## Image acquisition of larval and pupal VNCs after smFISH

Images were acquired on an Aberrior Infinity Line confocal microscope with Olympus UPLSAPO60XS, 60x/NA1.3 silicone oil objective. Image stacks were taken with the following settings: voxel size 50 µm × 50 µm (*XY*) and 150 µm (**Z**), pixel dwell time 5 µs, sequential line scanning, pinhole 0.93 airy unit. Parameters for the four channels used in this study are: excitation 485 nm, laser 30%, detection 495–550 nm, line accumulation 5; excitation 561 nm, laser power 30%, detection 571–630 nm, line accumulation 5; excitation 640 nm, laser power 15%, detection 650–695 nm, line accumulation 2; excitation 700 nm, laser power 60%, detection 710–780 nm, line accumulation 5.

## smFISH analysis

Each Lin A/15 cell was segmented in 3D in ImageJ/Fiji (*Rueden et al., 2017*; *Schindelin et al., 2012*; *Schneider et al., 2012*), using the LimeSeg plugin (*Machado et al., 2019*), on the GFP channel, with the following parameters: D_0~4–6, F pressure = 0.01, Z_scale = 3, range in d0 units ~4–6, number of integration step = −1, real *XY* pixel size = 50. For subsequent analysis, each segmented cell was exported into a separate ply file which was then imported in Matlab as a point cloud (The Math Works, Inc). The original stacks were imported in Matlab using the Bio-Formats toolbox (https://www.open-microscopy.org/bio-formats/downloads/). These stacks were then cropped around each cell using the point clouds generated by individual cell segmentation with LimeSeg.

   The mRNA spots were detected in 3D, in the mRNA channel of these cropped stacks, using the method and scripts described in *Raj et al., 2008*. In short, the spots were identified computationally by running a Matlab image processing script that runs the raw data through a filter (Laplacian of a Gaussian) designed to enhance spots of the correct size and shape while removing the slowly varying background. The filtered stacks are then thresholded to disregard remaining background noise. In order to choose an optimal threshold, all possible thresholds are computed. The thresholds were always chosen manually and close to the plateau. A 'check File stack' for each cell was generated in order to visualize the accuracy of the spot detection for a given threshold. In most of our samples, common thresholds were chosen for all the samples of a time point. However, specific threshold were occasionally chosen to give the best visual detection of mRNA spots in our datasets check files.

## PCCD method

The PCCD method, as described in our previous work (*Guan et al., 2022c*), aims to link the expression of a given TF to the birth order of an iMN by using the correlation between the birth order of iMNs and their spatial organization. In our Lin A/15 model, the EdU experiments reveal a good correlation

between the birth order of iMNs and their spatial distance from the NB in third instar larvae: young born iMNs are farther away from the NB compared to older iMNs. The final goal of this method is to predict the TF code expression pattern in each iMN in a third instar larva.

The method followed a series of steps:

### Step 1

From the imaging, assign spatial *x*, *y*, *z* coordinates and the expression (on/off) of a given TF to each Lin A/15 cell ($N > 15$, number of Lin A/15 immunostained for a given TF).

### Step 2

Calculate the Euclidean distance between the NB and the *x*, *y*, *z* coordinates of each iMN (relative distance).

### Step 3

Order iMNs in each Lin A/15 according to their distance to NB. This presents each Lin A/15 as an ordered sequence of iMNs (this defines the *x* axis position where cell #1 is defined as the furthest from the NB, i.e. the oldest iMNs on average). Then calculate the frequency of expression of all TFs as a function of their rank in each ordered Lin A sequence.

### Step 4

Apply a filter (Savitzky–Golay) to smooth each distribution.

### Step 5

Define the position in the sequence of the positive cell cluster(s) by using a peak detection method. Determine its length (average number of cells expressing a given TF in all Lin A/15 samples analyzed). Then find the position of the positive cell cluster with this average length compatible with the smoothed TF distribution. The position and its length are represented by a horizontal line.

### Step 6

Assemble all positive cell clusters for each TF on the same graph to reveal combinatorial TF code for each iMN. Convert the *x'* axis to a birth order axis (1–29) since the distance between iMN and the NB is tightly linked to their birth order. Define the coverage index at the border of all cell clusters.

## More details about the method include

Frequency histograms (Step 3): The frequency histograms of positive cells can in principle be computed in an either global or relative manner. Global means that at each position *i*, the observed number of positive cells $Pi$ will be normalized by the total number *N* of observed sequences for this TF. Thus, the frequency at rank *i* is defined as $Pi/N$. In contrast in the relative definition of frequencies, at each position *i*, the frequency is determined by the number of positive cells $Pi$ divided by the number of sequences $Ni$ for which this position has been measured, leading to a frequency at rank *i* defined as $Pi/Ni$. The relative measure avoids bias that possibly arises in the global method by considering as negative (by default) cells that are not observed in sequences that are too short to reach this index. In the sequel, we use this relative frequency histogram to limit such bias as much as possible.

Savitzky–Golay filter (Step 4): The Savitzky–Golay algorithm (polynomial filter) was set with a window of size 11 and a polynomial order of 3 (see scipy.signal.savgol_filter function from the scipy python library).

Peak detection method (Step 5): Peaks were detected as local maxima in the normalized TF distributions. Local maxima were determined according to local conditions. They had a minimal height (h_min = 0.2), a minimal distance from other peaks (d_min = 8), and a minimal prominence (p_min = 0.07). The prominence of a peak measures how much a peak is emerging clearly locally in the signal. It is defined as the vertical distance between the peak and the altitude of the largest region it dominates. These values were found to yield best peak interpretations over the whole set of TFs, in particular to detect multiple peaks in TF distributions such as RunxA. We used the function scipy.signal.find_peaks of the scipy library. In addition to the location of the different peaks *p* in the signal,

the whole range of *x* values (i.e. the *x* axis) is split in intervals [*ip,jp*] where each peak is prevailing. Cells contributing to peak *p* can thus only be found in the interval [*ip,jp*] for each peak *p*.

Positive cell clusters (Step 5): For each TF, the average number '*n*' of positive cells was computed in each Lin A/15 iMN observed Cluster corresponding to a detected peak *p* (using the span [*ip,jp*] as described above). The procedure varied according to whether only one peak was detected or more than one multiple peaks can be detected depending on the nature of the data and the parameters defining peaks (see above).

Case of a single detected peak (e.g. Jim): The span [*ip',jp'*] of the active cells under the peak *p* was computed within the span [*ip,jp*] by finding the horizontal span under the peak that extends exactly over *n* cells (green lines on the figures). This cluster [*ip',jp'*] of positive cells was assumed to correspond to all the cells expressing the TF.

Case of multiple detected peaks (e.g. Nvy (2 peaks) or RunxA (2 peaks)): The sequence was split into the regions [*ip,jp*] defined by each peak. Then the average number of positive cells '*n*1', '*n*2', …, are computed for each of the peak regions. Then the method proceeds within each region and its average number of positive cells as in the case of a single detected peak. This determines both the estimated length and the position of multiple positive cell clusters.

## Materials availability
All fly stocks used in this study are available from the corresponding author without restriction.

## Acknowledgements
We thank Alain Vincent, Filipe Pinto-Teixeira, and Cédric Maurange for comments on the manuscript. We thank the IGFL microscopy platform. This work was funded by AFM (#21999) and ANR-20-CE16-0007 DevandMaintain to JE. The PhD stipend of ZN is funded by the China Scholarship Council. This work was supported by the EquipEx+ Spatial-Cell-ID under the 'Investissements d'avenir' program (ANR-21-ESRE-00016). We acknowledge the contribution of SFR Biosciences (UAR3444/CNRS, US8/Inserm, ENS de Lyon, UCBL): Arthro-tool facility, for fly food preparation; as well as LYMIC – PLATIM, for support in microscopy. We also acknowledge the contribution of the IGFL Microscopy platform.

## Additional information

### Funding

| Funder | Grant reference number | Author |
|---|---|---|
| Agence Nationale de la Recherche | ANR-20-CE16-0007 DevandMaintain | Jonathan Enriquez |
| Agence Nationale de la Recherche | ANR-21-ESRE-00016 Spatial-Cell-ID | Jonathan Enriquez |
| China Scholarship Council | | Ziyan Nie |
| AFM Téléthon | AFM (#21999) | Jonathan Enriquez |

The funders had no role in study design, data collection and interpretation, or the decision to submit the work for publication.

### Author contributions
Wenyue Guan, Conceptualization, Formal analysis, Investigation, Writing - original draft, Writing - review and editing; Ziyan Nie, Formal analysis, Investigation, Visualization, Methodology, Writing - review and editing; Anne Laurençon, Resources, Validation, Investigation, Methodology, Writing - review and editing; Mathilde Bouchet, Software, Formal analysis, Supervision, Investigation, Visualization, Methodology, Project administration; Christophe Godin, Software, Formal analysis, Visualization, Methodology; Chérif Kabir, Software; Aurelien Darnas, Investigation; Jonathan Enriquez, Conceptualization, Supervision, Funding acquisition, Investigation, Methodology, Writing - original draft, Project administration, Writing - review and editing

## Author ORCIDs
Wenyue Guan http://orcid.org/0000-0002-9544-7563
Ziyan Nie http://orcid.org/0000-0001-9040-644X
Anne Laurençon https://orcid.org/0000-0003-1791-1882
Jonathan Enriquez https://orcid.org/0000-0002-7679-4674

Reviewer #1 (Public Review): https://doi.org/10.7554/eLife.91634.3.sa1
Reviewer #2 (Public Review): https://doi.org/10.7554/eLife.91634.3.sa2
Reviewer #3 (Public Review): https://doi.org/10.7554/eLife.91634.3.sa3
Author response https://doi.org/10.7554/eLife.91634.3.sa4

## Additional files

### Supplementary files
• Supplementary file 1. smFISH primary probe sequences for *Imp*, *Syp*, and *Dpn*. Sequences of the primary probes recognizing exonal sequences of *Imp*, *Syp*, and *Dpn*, designed using the Biosearch Technologies stellaris RNA FISH probe designer tool (free with registration, https://biosearchtech.com).

• MDAR checklist

### Data availability
All data are available within the article and its supplementary information. The PCCD method custom Python script can be downloaded at https://github.com/Wenyue2022/PCCD-note-book (copy archived at *Guan, 2022b*).The custom MATLAB script for smFISH analysis can be downloaded at https://github.com/Wenyue2022/smFish-analysis-source-code (copy archived at *Guan, 2022a*).

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
