## [Editor Report · eLife assessment]

Guan and colleagues present **solid** arguments to address the question of how a single neural stem cell produces a defined number of progeny, and what influences its decommissioning. The focus of the experiments are two well-studied RNA-binding proteins: Imp and Syp. This is **valuable** work that will be of interest to the scientific community.

---

## [Referee Report · Reviewer #1 (Public Review)]

This study addresses the temporal patterning of a specific *Drosophila* CNS neuroblast lineage, focusing on its larval development. They find that a temporal cascade, involving the Imp and Syb genes changes the fate of one daughter cell/branch, from glioblast (GB) to programmed cell death (PCD), as well as gates the decommissioning of the NB at the end of neurogenesis.

---

## [Referee Report · Reviewer #2 (Public Review)]

Guan and colleagues address the question of how a single neuroblast produces a defined number of progeny, and what influences its decommissioning. The focus of the experiments are two well-studied RNA-binding proteins: Imp and Syp. The Authors find that these factors play an important role in determining the number of neurons in their preferred model system of VNC motor neurons coming from a single lineage (LinA/15) by separate functions taking place at specific stages of development of this lineage: influencing the life-span of the LinA neuroblast to control its timely decommissioning and functioning in the Late-born post-mitotic neurons to influence cell death after the appropriate number of progeny is generated. The post-mitotic role of Imp/Syp in regulating programmed-cell death (PCD) is also correlated with a specific code of key transcription factors that are suspected to influence neuronal identity, linking the fate of neuronal survival with its specification. This paper addresses a wide scope of phenotypes related to the same factors, thus providing an intriguing demonstration of how the nervous system is constructed by context-specific changes in key developmental regulators. The bulk of conclusions drawn by the authors are supported by careful experimental evidence, and the findings are a useful addition to an important topic in developmental neuroscience.

---

## [Referee Report · Reviewer #3 (Public Review)]

This study by Guan and co-workers focuses on a model neuronal lineage in the developing *Drosophila* nervous system, revealing interesting aspects about: (a) the generation of supernumerary cells, later destined for apoptosis; and, (b) new insights into the mechanisms that regulate this process. The two RNA-binding proteins, Imp and Syp, are shown to be expressed in temporally largely complementary patterns, their expression defining early vs later born neurons in this lineage, and thus also regulating the apoptotic elimination. Moreover, neuronal 'fate' transcription factors that are downstream of Imp and signatures of early-born neurons, can also be sufficient to convert later born cells to an earlier 'fate', including survival. The authors provide solid evidence for most of their statements, including the temporal windows during which the early and the later-born motoneurons are generated by this model lineage, how this relates to patterns of cell death by apoptosis and that mis-expression of early-born transcription factors in later-born cells can be sufficient to block apoptosis (part of, and perhaps indicative of the late-born identity). Other studies have previously outlined analogous, mutually antagonistic roles for Imp and Syp during nervous system development in *Drosophila*, in different parts and at different stages, with which the working model of this study aligns. Overall, this study adds to and extends current working models and evidence on the developmental mechanisms that underlie temporal cell fate decisions.

---

## [Author Response]

The following is the authors’ response to the original reviews.

**Reviewer #1 (Public Review):**
This study addresses the temporal patterning of a specific *Drosophila* CNS neuroblast lineage, focusing on its larval development. They find that a temporal cascade, involving the Imp and Syb genes changes the fate of one daughter cell/branch, from glioblast (GB) to programmed cell death (PCD), as well as gates the decommissioning of the NB at the end of neurogenesis.

I believe there are some inaccuracies in this summary. We address temporal patterning during larval and pupal stages until the adult stage. The Imp and Syp genes change the fate of one daughter cell/branch from survival to programmed cell death (PCD). The change from glioblast (GB) to PCD, which occurs at an early time point, is not addressed here. The main point of the paper is missing:

• Last-born MNs undergo apoptosis due to their failure to express a functional TF code, and this code is post-transcriptionally regulated by the opposite expression of Imp and Syp in immature MNs.

**Reviewer #2 (Public Review):**
Summary:Guan and colleagues address the question of how a single neuroblast produces a defined number of progeny, and what influences its decommissioning. The focus of the experiments are two well-studied RNA-binding proteins: Imp and Syp. The Authors find that these factors play an important role in determining the number of neurons in their preferred model system of VNC motor neurons coming from a single lineage (LinA/15) by separate functions taking place at specific stages of development of this lineage: influencing the life-span of the LinA neuroblast to control its timely decommissioning and functioning in the Late-born post-mitotic neurons to influence cell death after the appropriate number of progeny is generated. The post-mitotic role of Imp/Syp in regulating programmed-cell death (PCD) is also correlated with a specific code of key transcription factors that are suspected to influence neuronal identity, linking the fate of neuronal survival with its specification. This paper addresses a wide scope of phenotypes related to the same factors, thus providing an intriguing demonstration of how the nervous system is constructed by context-specific changes in key developmental regulators.The bulk of conclusions drawn by the authors are supported by careful experimental evidence, and the findings are a useful addition to an important topic in developmental neuroscience.

I cannot summarize better the paper.

Strengths:A major strength is the use of a genetic labeling tool that allows the authors to specifically analyze and manipulate one neuronal lineage. This allows for simultaneous study of both the progenitors and post-mitotic progeny. As a result the paper conveys a lot of useful information for this particular neuronal lineage. Furthermore addressing the association of cell fate specification, taking advantage of this lab's extensive prior work in the system, with developmentally-regulated programmed celldeath is an important contribution to the field.Beyond Imp/Syp, additional characterization of this model system is provided in characterizing a previously unrecognized death of a hemilineage in early-born neurons.

Thanks!

Weaknesses:The main observations that distinguish this study from others that have investigated Imp/Syp in the fly nervous system is the role played in late-born post-mitotic neurons to regulate programmed cell death. This is an important and plausible (based on the presented findings) newly discovered role for these proteins. However the precision of experiments is not particularly strong, which limits the authors claims. The genetic strategy used to manipulate Imp/Syp or the TF code appears to be done throughout the entire lineage, or all neuronal progeny, and not restricted to only the late born cells. Can the authors rule out survival of the early born hemi-lineage normally fated to die? Therefore statements such as this:To further investigate this possibility, we used the MARCM technique to change the TF code of lastborn MNs without affecting the expression of Imp and Syp should be qualified to specify that the result is obtained by misexpressing these factors throughout the entire lineage.

We agree that our genetic manipulations affect the entire lineage or all neuronal progeny. We do not have genetic tools to gain such precision. We have changed our descriptions to specify the entire lineage or all neuronal progeny. As the reviewer raised, we were also concerned about the possibility that the overexpression of Imp or knockdown of Syp could induce the survival of the early-born hemilineage. We have two experiments that rule out this possibility:

(1) In late LL3 larvae, Imp OE or syp MARCM clones do not change the number of cells in LL3 larvae (see Guan et al., 2022), indicating that the hemilineage that died by PCD is not affected. If Imp or Syp played a role in the survival of the hemilineage, we would see at least a 50% increase in the number of MNs at this stage.

(2) The MARCM experiment using the VGlut driver to overexpress P35 or Imp allows us to manipulate only elav+ VGlut+ neurons. The hemilineage removed by PCD is elav- VGlut- and is not affected by this experiment. Consequently, the increase in MNs in adults with genetic manipulation can only be the result of the survival of the other hemilineage (elav+, VGlut+). Moreover, this experiment shows an increase in the number of neurons in the adult but not in LL3, demonstrating that the hemilineage (elav- VGlut-) is still removed by PCD with this genetic manipulation.

The authors make an observation that differs from other systems in which Imp/Syp have been studied: that the expression of the two proteins appears to be independent and not influenced by cross-regulation. However there is a lack of investigation as to what effect this may have on how Imp/Syp regulate temporal identity. A key implication of the previously observed cross-regulation in the fly mushroom body is that the ratio of Imp/Syp could change over the life of the NB which would permit different neuronal identities. Without cross-regulation, do the authors still observe a gradient in the expression pattern of time? Because the data is presented with Imp and Syp stained in different brain samples, and without quantification across different stages, this is unclear. The authors use the term 'gradient' but changes in levels of these factors are not evident from the presented data.

We have now quantified the transcriptional activity of Imp and Syp in the NB over time using smFISH. We have also quantified the relative expression of Imp and Syp protein in the NB over time by co-immunostaining. Additionally, we quantified the relative expression of Imp and Syp protein in postmitotic neurons as a function of their birth order in late LL3 larvae. All these data show an opposite temporal gradient of Imp and Syp in the NB and an opposite spatial gradient in immature neurons according to their birth order (Figure. 4). How these gradients are established in our system remains to be elucidated.

**Reviewer #3 (Public Review):**
This study by Guan and co-workers focuses on a model neuronal lineage in the developing *Drosophila* nervous system, revealing interesting aspects about: (a) the generation of supernumerary cells, later destined for apoptosis; and, (b) new insights into the mechanisms that regulate this process. The two RNA-binding proteins, Imp and Syp, are shown to be expressed in temporally largely complementary patterns, their expression defining early vs later born neurons in this lineage, and thus also regulating the apoptotic elimination. Moreover, neuronal 'fate' transcription factors that are downstream of Imp and signatures of early-born neurons, can also be sufficient to convert later born cells to an earlier 'fate', including survival.The authors provide solid evidence for most of their statements, including the temporal windows during which the early and the later-born motoneurons are generated by this model lineage, how this relates to patterns of cell death by apoptosis and that mis-expression of early-born transcription factors in later-born cells can be sufficient to block apoptosis (part of, and perhaps indicative of the late-born identity).Other studies have previously outlined analogous, mutually antagonistic roles for Imp and Syp during nervous system development in *Drosophila*, in different parts and at different stages, with which the working model of this study aligns.Overall, this study adds to and extends current working models and evidence on the developmental mechanisms that underlie temporal cell fate decisions.

I cannot summarize better the paper.

**Reviewer #1 (Recommendations For The Authors):**
While this is an interesting topic, I raised two issues in my original review.(1) Against the backdrop of numerous previous studies linking many developmental regulators, including tTFs, to programmed cell death in the developing CNS, which in several cases have involved identifying key PCD genes and decoding the molecular regulatory interplay between regulators and PCD genes, this study does not provide any new insight into the regulation of developmental PCD in the CNS.The authors have not added any new data to address this shortcoming.

I agree with the reviewer that we did not attempt to link Imp/Syp with the temporal transcription factor (tTF) cascade or spatial selectors such as Hox genes. However, this decision was intentional as our primary focus was on studying immature MNs. It is worth noting that the decommissioning of NBs by autophagic cell death or terminal differentiation, which is mediated by Imp/Syp in other lineages, has not been correlated with tTFs or spatial selectors. Although we have not directly examined the involvement of the hb + sv > kr > pdm > cas > cas-svp > Grh cascade in the decommissioning of the Lin A neuroblast, our preliminary data indicate that Hb, Sv, Pdm, and Cas are not expressed in the Lin A NB, while Grh is consistently expressed in the NB (Wenyue et al., 2022). Thus, it is less likely that this particular tTF cascade is not implicated in Lin A neuroblast decommissioning. In contrast, spatial selectors, such as the Hox gene Antp, play an opposing role compared to HOX transcription factors in abdominal NBs. In the Lin A lineage, Antp promotes survival (Baek, Enriquez, & Mann, 2013). Here, to avoid repeating what has already been described in the literature, we focused on the role of Imp/Syp in postmitotic neurons and revealed that the precise elimination of MNs is linked to the control of TFs expressed in the MNs.

(2) I raised the issue that it is unclear if Imp/Syp acts in the NB, and/or in IMC/GMC, and/or in the daughter cells generated from these.

I agree with the reviewer's concern regarding the unclear function of Imp/Syp, i.e., whether it acts in the NB, IMC/GMC, or daughter cells. To address this, one possible approach would be to attempt rescuing Imp and Syp mutants by transgenic expression in specific cell types, such as NBs, IMC/GMC, or GB/daughter cells. However, we have not conducted such experiments as we were skeptical about the outcome. Previous published work has used drivers expressed in NBs, IMC/GMC, or postmitotic neurons to decipher the function of a gene in a specific cell type. But the results of these experiments must be taken with caution. Using NB/GMC drivers to study gene function can lead to effects not only in the NB but also in its progeny, including GMC or postmitotic neurons, due to the perdurance and stability of the Gal4 and UAS-gene expression system. For instance, dpn-Gal4 UASGFP not only labels the NB but also many of its progeny, even if Dpn is only expressed in NBs. And elav-Gal4 is expressed in the NB and GMCs.

However, our overexpression of Imp in immature neurons using Vglut demonstrates that Imp promotes cell survival through an autonomous function in these neurons. This driver is only expressed in postmitotic neurons (elav+) and not in the NB, IMC/GMC, or in the hemilineage eliminated by cell death (elav-vglut-).

**Reviewer #2 (Recommendations For The Authors):**
Oddly knockdown of Imp in the neuroblast (Fig. 5D) only led to death at 8h APF, when Imp is no longer expressed. Do the authors have an explanation as to how the stem cell can survive until this point? A discussion would be helpful.

The simple explanation is the efficiency of RNAi. The imp-/- MARCM clones (Guan et al., 2022) lead to a stronger reduction of MNs in LL3.

A simple experiment I would recommend is to repeat the antibody stainings of staged larvae/pupae (Fig. 4) having the anti-Imp/Syp antibodies in the same brain sample, and perhaps a quantification of the ratio in the NB. Given the species in which the ABs were raised seem compatible, this should be feasible. As it stands now, there is no indication of whether the ratio of Imp vs Syp change over time.

We have now quantified the transcriptional activity of Imp and Syp in the NB over time. We have also quantified the relative expression of Imp and Syp proteins in the NB over time and quantified the relative expression of Imp and Syp proteins in postmitotic neurons as a function of their birth in late LL3 larvae. How these gradients are established in our system still remains to be

Minor errors/suggestions:Fig 4. Time legend at the top goes A, B, C, E, F (no D). So it doesn't match the panels below

Yes, we have made the corrections.

Sentence repeated in Intro:The process of terminating NB neurogenesis through autophagic cell death or terminal differentiation is commonly referred to as decommissioning.

Yes, corrections have been made.

IN FIGURE 1 THEY SAY 'TYPE IB' AND IN FIGURE 2 THEY SAY 'TYPE 1B'

We have changed it to type 1b.

In Fig2A-It's hard to see lack of Elav and Fig2G-It's hard to see presence of Dcp1. Panels could be adjusted to emphasize these results

We have increased the size of the panels and made two separate panels where only the elav and Dcp1 signals are present.

Observations that the result is equivalent in all thoracic segments is expected, since all legs need the same number of neurons. This is nice to have but can be in the supplement.Overall the figure number seems excessive, especially considering much of the results included (particularly the NB results) are findings consistent with previous papers and some is characterization of the system that does not fit well with the main focus regarding Imp/Syp (i.e death of one hemi-lineage):Figure 5 and 6 can be joined as one.

We have combined Figures 5 and 6, showing only the T1 segments.

There is some discrepancy between graphs Fig7F and K: At LL3 the number of neurons is different for the control in 7F and the count in K

Yes, because the genetic backgrounds are not the same and we are not counting the same type of cells. In 7F, we are counting the elav+ and VGlut+ cells, whereas in Figure 7K, we are counting all the elav+ in Lin A, including those elav+ VGlut-. VGlut expression arrives a bit later after elav+, which is why we have fewer elav+ cells in 7F. In other words, VGlut MARCM clones do not label all Lin A elav+ cells. I have clarified this in the figure.

**Reviewer #3 (Recommendations For The Authors):**
Main comment: on the notion of Imp and Syp gradients:p. 5, related to figure 4 - there are clearly distinct windows for predominantly (if not exclusively) Imp, and later, Syp expression in lineage 15, with a phase of co-expression.However, based on the data shown, it is unclear whether these windows represent gradients, as repeatedly stated. If the notion of gradients is derived from other studies, on other lineages, then this would be good to clarify. Alternatively, the idea of temporally opposing gradients of Imp and Syp would need to be demonstrated for this lineage.For example, a more accurate way to describe this study's data is given on p.7 "In conclusion, our findings demonstrate that the opposite expression pattern of Imp and Syp in postmitotic neurons precisely shapes the size of Lin A/15 lineage by controlling the pattern of PCD in immature MNs (Fig. 8)."

We have now quantified the transcriptional activity of Imp and Syp in the NB over time. We have also quantified the relative expression of Imp and Syp proteins in the NB over time. We have also quantified the relative expression of Imp and Syp proteins in postmitotic neurons as a function of their birth in late LL3 larvae. How these gradients are established in our system still remains to be identified.

Minor points:p.6, related to figure 7: Are numbers of EDU- early born and EDU+, late born, MNs expressed as means in the main text? As written, it suggests absence of any variability, which one would expect and which is shown in Fig.7 data.

Yes, we have added averages in the text.

Methods: the author name 'Lacin' has been mis-spelled

Sorry about that, it's been corrected.